# FROM FIELDS TO RANDOM TREES

**Yaomin Wang** [1,2] , **Xiaodong Luo**[2] , **Tianshu Yu**[1]*
[1] School of Data Science, The Chinese University of Hong Kong, Shenzhen
[2] Shenzhen Research Institute of Big Data
`yaominwang@link.cuhk.edu.cn`
`{xiaodongluo, yutianshu}@cuhk.edu.cn`

## ABSTRACT

This study introduces a novel method for performing Maximum A Posteriori (MAP) estimation on Markov Random Fields (MRFs) that are defined on locally and sparsely connected graphs, broadly existing in real-world applications. We address this long-standing challenge by sampling uniform random spanning trees (SPT) from the associated graph. Such a sampling procedure effectively breaks the cycles and decomposes the original MAP inference problem into overlapping sub-problems on trees, which can be solved exactly and efficiently. We demonstrate the effectiveness of our approach on various types of graphical models, including grids, cellular/cell networks, and Erdős-Rényi graphs. Our algorithm outperforms various baselines on synthetic, UAI inference competition, and real-world PCI problems, specifically in cases involving locally and sparsely connected graphs. Furthermore, our method achieves comparable results to these methods in other scenarios. The code of our model can be accessed at `https://github.com/LOGO-CUHKSZ/From-fields-to-random-trees.git`.

## 1 INTRODUCTION

In this paper, we investigate a novel approach to infer on Markov Random Fields defined over *sparsely and locally connected graphs* via random spanning tree sampling. Formally, an MRF is defined over an undirected graph $\mathcal{G} = (\mathcal{V}, \mathcal{E})$, where $\mathcal{V}$ represents the index set of random variables and $\mathcal{E}$ corresponds to the edge set implying dependencies among these variables. Each random variable in $x_i \in \mathcal{X}$ takes a value from a finite alphabet $\mathcal{X}$ ($x_i$ can be a vector) with node index $i \in \mathcal{V}$, and the joint distribution is expressed as a product of potential functions, each associated with a subset of variables that forms a clique in $\mathcal{G}$. In this paper, we consider minimizing energy functions with unary terms $\theta_i$ and pairwise terms $\theta_{ij}$, then the MAP estimate is:

$$\min_X E(X) = \min_{\{x_i\}, \forall i \in \mathcal{V}} \left\{ \sum_{i \in \mathcal{V}} \theta_i(x_i) + \sum_{(i,j) \in \mathcal{E}} \theta_{ij}(x_i, x_j) \right\}. \tag{1}$$

where $X = \{x_i\}_{i \in \mathcal{V}}$. Optimization problem over MRFs expressed in equation 1 is a prevalent model in the realm of probabilistic graphical modeling (Clifford & Hammersley, 1971). Notably, MRF and its counterparts enable the principled and simplified representation of joint probability distributions over a set of variables, conditioned upon the structure of an undirected graph. The intrinsic capability of MRFs to model contextual relationships in data offers a cohesive and efficient framework for addressing inference and estimation challenges across a diverse array of scientific and engineering disciplines, including computer vision (Wang et al., 2013; Su et al., 2021), 5G networks (Kumar et al., 2022), pathology image analysis (Li et al., 2020) and in other fields combining the power of GNN (Xu et al., 2021a; Wu et al., 2020).

The problem in equation 1 is known to be NP-hard in general. Early trials to efficiently and suboptimally solve equation 1 dates back to the 80's of the previous century, when Judea Pearl developed the prestigious Belief Propagation (BP) algorithm (Pearl, 1988). Motivated by this breakthrough, a series of BP variants were further proposed (Wiegerinck & Heskes, 2002; Yedidia et al., 2005; Montanari & Rizzo, 2005). Other prominent methods include Mean Field Approximation (Saito et al., 2012; Zhang, 1993; Zhang & Hanauer, 1995), Graph Cuts (Greig et al., 1989), and Junction Trees (Aji & McEliece, 2000), to name a few. These methods offer varying trade-offs between accuracy and computational efficiency. To the best of our knowledge,

---

*Corresponding author.

no single method stands out as state-of-the-art over all existing problems spanning various scales, topologies, and problem distribution.

In this study, we put our special focus on sparsely and locally connected graphs. In realistic scenarios, such topologies broadly exist in power grid (Cuffe & Keane, 2017), 5G networks (Agiwal et al., 2016; Liu & Zou, 2020), transportation networks (Yunfei Ma & Razavi, 2022), and even social networks (Majeed et al., 2020). Efficient MAP inference in the corresponding applications is crucial. Beyond our focus, we emphasize that our study can be flexibly extended into other types of graphs.

Diverging from existing well-established strategies, our approach leverages the structure of random spanning trees to efficiently infer on MRFs. In a nutshell, our approach consists of "sampling random trees – solving MRFs on trees – merging" steps. Concretely, instead of solving the MRF on the entire graph directly, we sample multiple spanning trees from the original graph and solve the MRF independently on each tree, on which exact inference is tractable. The final solution to equation 1 is then approximated by merging the solutions from all sampled trees. Our approach thus combines the benefits of exact tree-based inference procedures with the flexibility of sampling methods, creating a balance between computational efficiency and accuracy.

We conduct experiments on synthetic instances, UAI competition instances (uai), and real-world Physical Cell Identity (PCI) instances, covering a variety of problem scales, types of topologies, and energy functions. We observe superior performance against several baselines in the sparse and local network setting, and comparable performance in other settings, which strongly pose the promise of the proposed method in various scenarios.

In summary, our contribution through this proposed method introduces a scalable and efficient sampling-based approach to infer MRFs on locally and sparsely connected graphs, mitigating the drawbacks of existing methods. Superior performance is observed in benchmarks across a wide spectrum. By transcending the traditional methodologies and introducing structural simplification via topological sampling over trees, this work posits rich potential to solve intricate MRF problems in the real world.

## 2 RELATED WORKS

In MRFs, the energy function is linked to a graph-structured probability distribution. A significant inference challenge in MRFs is determining the MAP configuration. Although minimizing the energy function of MRF models is NP-hard, advances in inference techniques have significantly expanded the model's capabilities. The success in solving the MAP estimation problem on cycle-free graphs is highly dependent on the graph's structure. In these graphs, the MAP problem can be effectively tackled using a variant of the min-sum algorithm (Clifford & Hammersley, 1971; Besag, 1974; Kumar et al., 2005), which facilitates message passing between nodes and serves as an extension of the Viterbi algorithm (Yedidia et al., 2003) to arbitrary cycle-free graphs. For graphs containing cycles, graph cut methods (Komodakis et al., 2007; Roy & Cox, 1998; Boykov et al., 1998; Ishikawa & Geiger, 1998; Szummer et al., 2008; Ishikawa, 2003; Schlesinger & FLACH, 2006) offer a potent solution by employing min-cut/max-flow strategies to efficiently reduce discrete MRFs' energy.

The belief propagation (BP) algorithm, introduced by Pearl (Pearl, 1982; 1988) in 1982, is an efficient iterative inference algorithm for Bayesian belief networks, functioning through fixed-point message passing. Its adaptability has made it a widespread solution for various types of MRFs. Nonetheless, BP encounters difficulties with models containing loops. Loopy belief propagation (LBP) attempts to resolve this by iterating message passing within graphs, even with loop presence (Weiss & Freeman, 2001; Felzenszwalb & Huttenlocher, 2004; Frey & Mackay, 2002). While LBP has shown efficacy in several vision tasks (Sun et al., 2002), it does not ensure fixed-point convergence, and its theoretical underpinnings remain elusive. The quest for a flexible, convergence-guaranteed method persists, yet significant advancements have been made to enhance BP's performance. The method proposed by Grim & Felzenszwalb (2023) enhances BP by adjusting the significance of input messages through a discount factor for remote nodes in the message passing chain. Additionally, leveraging graph topology for decomposition mitigates the impact of loops. In Yan et al. (2023), BP's inefficiency in large-scale MRFs is addressed by constructing a hierarchical framework, facilitating inference via energy connections between layers. Another strategy, detailed by Hamze & de Freitas (2004), involves partitioning graphs into two segments to serve as mutual evidence for updates, although its applicability is limited to graphs with predefined structures. In Kirkley et al. (2021), they propose graph decomposition using primary circles of a specified length from any given node, aiming to circumvent short loop influences, yet it is not effective for large-scale graphs. Integrating tree structures to break loops within the graph, the Junction Tree Algorithm (JTA) (Aji & McEliece, 2000), an exact inference method for arbitrary graphs, entails finding a maximum spanning tree across the largest cliques of a triangulated graph, a task known to be NP-hard, thereby limiting its practicality. In the realm of pairwise MRFs, problems are formulated as integer linear programming (ILP) (Wainwright et al., 2005; Kolmogorov, 2006), where solutions are derived from a dual problem using a convex combination of trees. This class of algorithms, known as tree-reweighted message passing (TRW) techniques, encompasses edge-based (TRW-E) and tree-based (TRW-T) schemes, both of which lack guaranteed convergence, potentially looping infinitely. The sequential TRW-S scheme (Kolmogorov, 2006) achieves a state of weak tree agreement (WTA), ensuring the lower bound stabilizes, albeit requiring substantial time

to reach this stage. TRBP has long been considered the state-of-the-art (SOTA) methodology and has been adapted in Xu et al. (2021b) to exploit modern GPUs for accelerated inference processes.

Several approaches have leveraged tree structures to address graph-related challenges. Batra et al. (2010) proposes a graph decomposition method using outer-planar graphs. However, this approach is restricted to planar MRFs (such as grids or superpixel adjacency graphs), and the computational complexity in determining both the required number and structure of subgraphs limits its applicability to general and large-scale graphs. The method in Pletscher et al. (2009) employs tree log-likelihood for approximation, while Skurikhin (2014) combines log-likelihood maximization with gradient descent for inference, specifically on grid structures. Both methodologies rely on voting mechanisms for final predictions. Using trees and Non-parametric Belief Propagation (Savic & Zazo, 2010) improve the accuracy in solving the localization problem in communication networks under 100 nodes. None of these three methods implement message correction on edges, potentially leading to deviations from original messages, as demonstrated in Lemma 1. While Bradley & Guestrin (2010) addresses structure learning, our work specifically focuses on MAP inference in MRFs and CRFs. Additionally, their approach is limited to graphs exhibiting tree-like structures. Trees could also work on binary inference problems (Cesa-Bianchi et al., 2010) through graph transformation, converting tree structures into line graphs for prediction purposes. Our work proposes a novel methodology that addresses two critical challenges in MRFs: circumventing the computational complexities associated with loop structures while maintaining faithful approximations of the original problem formulation. This approach demonstrates particular efficacy in handling large-scale MRF instances and problems that can be formulated within the MRF framework.

## 3 Preliminaries

**Markov Random Field**. MRFs can be used to model probabilistic undirected graphs. We follow the notations in Section. 1 and further suppose $|\mathcal{E}| = M$ and $|\mathcal{V}| = N$. In this paper, each node $i$ corresponds to a discrete state variable $x_i \in \mathcal{X}, \forall i \in \mathcal{V}$, where $\mathcal{X}$ is a finite alphabet. And there is a conditional independence between variables

$$\mathbb{P}(x_i|X\backslash\{x_i\}) = \mathbb{P}(x_i|\{x_j\} \text{ for } (i,j) \in \mathcal{E}). \tag{2}$$

where $\mathbb{P}(\cdot)$ is the probability function throughout this paper. Thus, the distribution on the graph $\mathcal{G}$ can be factorized into a product of local Markov potentials

$$\mathbb{P}(X) = \frac{1}{Z}\exp(-E(X)) = \frac{1}{Z}\exp\left(-\sum_{i\in\mathcal{V}}\theta_i(x_i) - \sum_{(i,j)\in\mathcal{E}}\theta_{ij}(x_i, x_j)\right) \tag{3}$$

where $Z$ is the partition function, $\theta_i(\cdot)$ denotes the unary energies, $\theta_{ij}(\cdot)$ represent the pairwise interaction energies, and $E$ is the energy function. The specific forms of energy functions are determined by the nature of the problems and they are all known in most cases. In our study, we aim to maximize the posterior probability by finding the optimal configuration of the hidden variables $X := \{x_i\}$ (assign appropriate values to each of the random variables). This corresponds to minimizing the energy function:

$$X^{\text{opt}} = \arg\max_{X\in\mathcal{X}}\mathbb{P}(X) = \arg\min_{X\in\mathcal{X}} E(X) \tag{4}$$

**Sum-Product Belief Propagation**. The key subroutine of our Spanning Tree message passing algorithms is sum-product belief propagation of Pearl (Pearl, 1988). BP is an algorithm for approximate minimization of energy $E(x)$ as in equation 3; it is exact if the graph does not have loops (*e.g.*, trees and chains). Sum-product BP maintains a directional message $M_{ij}$ from node $i$ to node $j$. The basic operation of BP is to pass the message from node $i$ to node $j$ along the edge $(i,j)$. After receiving all the messages from node $i$'s neighbors, the marginal distribution of node $i$ reads:

$$\mathbb{P}(x_i|X\backslash\{x_i\}) = \mathbb{P}(x_i)\prod_{(i,j)\in\mathcal{E}} M_{ji} \tag{5}$$

where $\mathbb{P}(x_i)$ is the prior marginal of $x_i$. In the absence of specific prior information, we will assume that all priors are uniformly distributed. BP algorithm iteratively passes messages in a specific order until a stopping criterion is met. One numerical method that improves the performance of BP is to stabilize the fixed point iteration scheme with damping (Murphy et al., 2013), which helps prevent oscillations between two steady states. This method involves replacing the term $\prod_{(i,j)} M_{ij}$ in equation 5 with a convex combination of the received messages:

$$M_{ij}^t = (1-\alpha)M_{ij}^t + \alpha M_{ij}^{t-1} \tag{6}$$

where $\alpha \in (0,1)$ is known as the damping factor and $t$ refers to the iteration number.

**Random Spanning Trees**. Given a graph $\mathcal{G} = (\mathcal{V}, \mathcal{E})$, we denote $\mathcal{T}$ the collection of all the spanning trees $T$ in $\mathcal{G}$ and $\Omega(\mathcal{T})$ some distribution over $\mathcal{T}$. The spanning trees we select from $\mathcal{T}$ is $\{T_k \in \mathcal{T}|k \in \mathcal{K}\}$, where $\mathcal{K}$ is the index set of the trees. Denote $\rho_T$ the probability of sampling a spanning tree $T$ from $\mathcal{T}$. Throughout the paper, we assume that each distinct tree is sampled with equal probability, *i.e.*, for any two distinct spanning trees $T_1, T_2 \sim \Omega(\mathcal{T})$, $\rho_{T_1} = \rho_{T_2}$. For each edge $(i,j) \in \mathcal{E}$, we denote $\rho_{ij} = \mathbb{P}((i,j) \in T, T \sim \Omega(\mathcal{T}))$ the probability that an edge $(i,j)$ appears in a random spanning tree.

## 4 MAP INFERENCE USING TREE SAMPLING

In this section, we present our approach and examine the potential benefits of approximating the original problem by combining the results via the analysis of random spanning trees in the original graph. Furthermore, we present a heuristic that has the potential to improve the approximation outcome by capitalizing on the information obtained during the "BP – update – inference" procedure. Subsequently, we dive into a thorough examination of the underlying logic behind our algorithm and conduct a comprehensive analysis of its complexity.

---

**Algorithm 1:** MAP on Graph using Spanning Tree Sampling

---

**Input:** Graph $\mathcal{G}$, index set of spanning trees $\mathcal{K}$, number of iterations $iter$.
**Output:** overall states $X$.

1   $\mathcal{T}^{\text{cand}} \leftarrow \{T_k\}_{k=1}^{|\mathcal{K}|}$, for $T_k \sim \mathcal{T}$ ;           `// sample a candidate tree set`
2   **while** *not converge* **do**
3      **for** $k \in \mathcal{K}$ **do**
4         Apply BP to $T_k \in \mathcal{T}^{\text{cand}}$ to compute $p_{T_k}^t(x_i|X\backslash\{x_i\}) \,\forall x_i \in X$ ;
5      $\mathcal{M} = \{p(x_i|X\backslash\{x_i\})|x_i \in X\} = \{ \prod_{k\in\mathcal{K}} (p_{T_k}^t(x_i|X\backslash\{x_i\})|x_i \in X\}$ ;   `// update marginals`
6      $X^{\text{Gibbs}} = \text{GibbsSampler}(\mathcal{M})$;
7      $X^{\text{Greedy}} = \text{GreedySelector}(\mathcal{M})$;
8      $E_{temp}^{best,t} = \min(E_{temp}^{best,t-1}, E(X^{\text{Gibbs}}), E(X^{\text{Greedy}}))$;
9      Update $X_{temp}^{best,t}$ according to $E_{temp}^{best,t}$ ;           `// update best X`
10 $X = X_{temp}^{best,t}$;

---

### 4.1 ALGORITHM

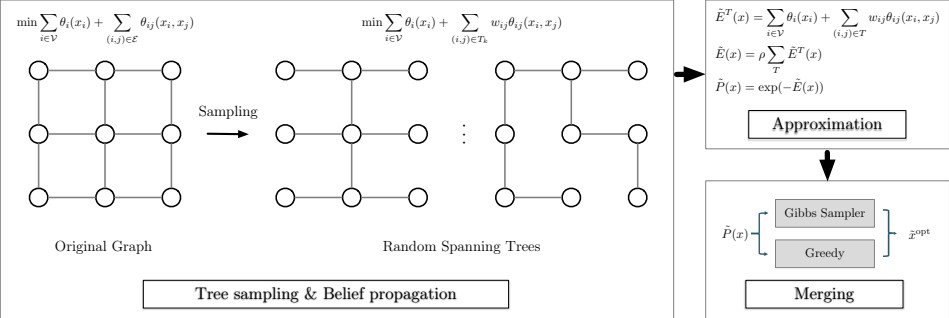

Figure 1: Pipeline of the proposed approach.

Procedures in the proposed approach are summarized in Algorithm 1 and a pipeline diagram is illustrated in Fig. 1. While the presence of loops in the original graph renders it analytically intractable, conducting belief propagation on each of the spanning trees efficiently delivers exact solutions. Note a spanning tree is a structure that can encompass all nodes and capture a significant amount of the relationships between nodes without loops, which maximally utilizes original information and avoids the oscillations.

By sampling spanning trees from the graph and doing belief propagation on these trees, we are actually trying to formulate a series of sub-problems of the original problem and solve the original problem by solving these sub-problems. The sub-problem on a spanning tree $T$ is shown below:

$$\min_X E^T(X) = \min_{\{x_i\}, \forall i \in \mathcal{V}} \left\{ \sum_{i\in\mathcal{V}} \theta_i(x_i) + \sum_{(i,j)\in\mathcal{T}} \theta_{ij}(x_i, x_j) \right\}. \tag{7}$$

To ensure that the combined energy of the spanning trees aligns with the energy of the original graph, and consequently that the joint distribution over the trees matches the joint distribution of the original graph, it is crucial to validate the correctness of the beliefs propagated within the trees. During the belief propagation procedure, messages passed along an edge represent the marginalized joint distribution of that edge, which inherently depends on the pairwise energy term.

To mitigate the bias introduced by the sampling procedure, we adjust the pairwise beliefs on each uniformly sampled spanning tree using the probability $\rho_{ij}$ of an edge $(i,j)$ appearing in a uniformly sampled spanning tree. The adjustment coefficient for the pairwise energy on edge $(i,j)$ is given by $w_{ij} = 1/\rho_{ij}$.

To accurately estimate the probability $\rho_{ij}$ of an edge appearing in a uniformly sampled spanning tree, we utilize the concept of *effective resistance*, as detailed in Section 4.3. The adjusted sub-problem on a spanning tree $T$ is formulated as follows:

$$\min_X E^T(X) = \min_{\{x_i\}, \forall i \in \mathcal{V}} \left\{ \sum_{i \in \mathcal{V}} \theta_i(x_i) + \sum_{(i,j) \in \mathcal{T}} w_{ij}\theta_{ij}(x_i, x_j) \right\}. \tag{8}$$

Then we could use the energy of each sampled tree to approximate the original energy of the graph:

$$\tilde{E}(X) = \sum_{k \in \mathcal{K}} \rho_{T_k} \left[ \sum_{i \in \mathcal{V}} \theta_i(x_i) + \sum_{(i,j) \in T_k} w_{ij}\theta_{ij}(x_i, x_j) \right] = \sum_{i \in \mathcal{V}} \theta_i(x_i) + \mathbb{E}_{T \sim \Omega(\mathcal{T})}[\tilde{\Theta}(X, X)], \tag{9}$$

where $\rho_{T_k}$ is the probability that $T_k$ is sampled and $\tilde{\Theta}(X, X)$ is the pairwise energy term after reweighted. And the second term of the right-hand side is the expectation of the adjustment pairwise term according to the distribution of the random uniform spanning trees. By Lemma 1, we could make sure the merged MAP problem is actually an approximation of the original problem. We use a toy example to show the necessity of weight adjustment in **Appendix I**.

After adjustment, the incoming message of vertex $i$ from vertex $j$ now becomes $\tilde{M}_{ji} = (M_{ji})^{w_{ij}}$.

Then equation 3 on a certain tree $T$ now becomes

$$\mathbb{P}^T(X) = \frac{1}{Z^T} \exp(-E^T(X)) = \frac{1}{Z^T} \exp\left( -\sum_{i \in \mathcal{V}} \theta_i(x_i) - \sum_{(i,j) \in T} w_{ij}\theta_{ij}(x_i, x_j) \right). \tag{10}$$

Now we could approximate the joint distribution via:

$$\tilde{\mathbb{P}}(X) = \prod_{k \in \mathcal{K}} \mathbb{P}_{T_k}(X)^{\rho_{T_k}} = \frac{1}{\tilde{Z}} \exp\left( -\sum_{k \in \mathcal{K}} \rho_{T_k} \left[ \sum_{i \in \mathcal{V}} \theta_i(x_i) + \sum_{(i,j) \in T_k} w_{ij}\theta_{ij}(x_i, x_j) \right] \right). \tag{11}$$

After applying belief propagation on sampled trees, we could obtain the marginals $p_{T_k}(x_i|X\backslash\{x_i\})$ on each tree. They are subsequently merged to approximate the true marginal distribution. For the initial marginals, without any specification, we will assume they are all uniform. To obtain the estimations, we do the following procedure to merge the marginals of each variables on each trees since the summation in energy is the multiplication when calculating probability:

$$\tilde{p}(x_i|X\backslash\{x_i\}) \propto \prod_{k \in \mathcal{K}} (p_{T_k^t}(x_i|X\backslash\{x_i\})). \tag{12}$$

We employed two methods, Gibbs sampling (Geman & Geman, 1984), and Greedy selection, to assign values to each of the variables based on the estimation of the marginal distributions as in Line 6 and 7 in Algorithm 1. These methods are implemented in modules named GibbsSampler and GreedySelector. The GibbsSampler samples a label for each variable based on the estimated marginal distribution, this label configuration is called $X^{Gibbs}$, while the GreedySelector selects the label with the highest value from the estimated marginal distribution, this label configuration is called $X^{Greedy}$. Since the sampling procedure will introduce uncertainty and greedy selection on nodes doesn't mean the combination is the best, we will keep recording the best configuration so far $X_{temp}^{best,t}$ during the iterations. We will update this configuration if some of the configuration given by the two modules yields the lowest energy so far $E_{temp}^{best,t}$. When the algorithm terminate, the best configuration that is found would be the final output.

**Complexity Analysis** can be found in **Appendix D**.

**Lemma 1.** *Given a uniform spanning tree distribution $\Omega(\mathcal{T})$ and the corresponding edge appearance probabilities $\{\rho_{ij}|\forall(i,j) \in \mathcal{E}\}$, when $|\mathcal{K}| = |\mathcal{T}|$, the approximation energy Eq. 9 and the original energy coincide.*

Since acquiring the true tree selection probability $\rho_T$ is intractable in practice, we use Monte Carlo Approximation to do the approximation when implementing the algorithm, by Lemma 2, you can see the condition in Lemma 1 still holds.

**Lemma 2.** *By applying the Monte Carlo Approximation, the expectation of the approximation is the original energy.*

$$\mathbb{E}_{T_k \sim \Omega(\mathcal{T})} \left[ \sum_{i \in \mathcal{X}} \theta_i(x_i) + \frac{1}{|\mathcal{K}|} \sum_{k \in \mathcal{K}} \sum_{(i,j) \in T_k} w_{ij}\theta_{i,j}(x_i, x_j) \right] = \sum_{i \in \mathcal{X}} \theta_i(x_i) + \sum_{(i,j) \in \mathcal{E}} \theta_{ij}(x_i, x_j) \tag{13}$$

**Theorem 1.** *Given spanning tree distribution* $\Omega(\mathcal{T})$ *and the corresponding edge appearance probability* $\{\rho_{ij} | \forall (i,j) \in \mathcal{E}\}$, *the following error bound of the approximation energy Eq. equation 9 holds with probability at least* $1 - \delta$.

$$|E(X) - \tilde{E}(X)| \leq \sqrt{\frac{1}{|\mathcal{K}|} \sum_{(i,j) \in \mathcal{E}} \theta_{ij}^2(x_i, x_j)(\frac{1 - \rho_{ij}}{\rho_{ij}})} \frac{1}{\sqrt{\delta}}. \tag{14}$$

*When*

$$|\mathcal{K}| \geq \frac{1}{\delta \eta^2} \sum_{(i,j) \in \mathcal{E}} \theta_{ij}^2(x_i, x_j)(\frac{1 - \rho_{ij}}{\rho_{ij}}), \tag{15}$$

*we have* $P(|E(X) - \tilde{E}(X)| \geq \eta) \leq \delta$.

The proof of Lemma 1, Lemma B and Theorem 1 can be found in Appendix A and Appendix C respectively. It is obvious that due to the term $(\frac{1 - \rho_{ij}}{\rho_{ij}})$, we can achieve good quality results with only a few trees when the graph is sparse.

The error bound exhibits an inverse relationship with the edge selection probability $\rho_{ij}$. As $\rho_{ij}$ approaches smaller values, the factor $(\frac{1 - \rho_{ij}}{\rho_{ij}})$ grows significantly, leading to a looser error bound. This relationship provides insight into the performance disparity between sparse and dense graphs. In sparse graphs, each edge typically has a higher probability of being included in a spanning tree, since fewer alternative paths exist between vertices. Conversely, in dense graphs, the abundance of potential paths results in lower individual edge selection probabilities. Consequently, when $\rho_{ij}$ is larger (as in sparse graphs), the term $(\frac{1 - \rho_{ij}}{\rho_{ij}})$ remains relatively small, yielding a tighter error bound and better estimation accuracy. Moreover, this theoretical analysis aligns with our empirical observations of superior performance on sparse graphs. This mathematical relationship explains why our method naturally performs better on sparse structures, where the higher edge selection probabilities contribute to more reliable estimates.

## 4.2 RANDOM UNIFORM SPANNING TREE

The key idea behind our algorithm is utilizing the partial information contained inside the original graphs. To uniformly sample a random spanning tree from the given graph, we employed the widely used Wilson's algorithm (Wilson, 1996). This algorithm, based on random walks and removal, ensures the uniformity of the resulting spanning tree selection. In general, the time complexity of Wilson's algorithm is $O(N^3)$. It could be significant when the size of the graph increases. Using methods such as Depth-First Search (DFS) would be more efficient, given the time complexity of $O(M + N)$. However, these methods make it challenging to identify the distribution of the trees. As a result, we cannot adjust the messages based on the edge selection probabilities. Therefore, one of the future trials is to incorporate advanced (approximate) sampling procedures such as Schild (2018) to reduce the time complexity.

## 4.3 EFFECTIVE RESISTANCE

We employ algorithms approximating effective resistance to obtain $\rho_{ij}$. Effective resistance is used in electrical network analysis and graph theory to measure the resistance between two vertices in a graph (Lyons & Peres, 2016) – how difficult it is for current to flow between two points in a network. The terminology of effective resistance originates from the following observation: Given the resistance on all edges, if one removes all vertices of $\mathcal{G}$ except $(s, t)$ and replaces the whole network with a resistance of resistor $\text{Reff}(s, t)$ between $(s, t)$, then, the energy (and the potential difference) of all electrical flows between $(s, t)$ remains invariant.

There is a strong connection between $\rho_{ij}$ and the corresponding effective resistance. By applying Kirchhoff's effective resistance formula (Lyons & Peres, 2016), we can establish the following lemma.

**Lemma 3** ( (Madry et al., 2014; Doyle & Snell, 2000)). *For any unweighted graph* $\mathcal{G} = (\mathcal{V}, \mathcal{E})$, *any edge* $(i, j) \in \mathcal{E}$, $\rho_{ij} = \mathbb{P}[(i, j) \in T, T \sim \Omega(\mathcal{T})] = \text{Reff}(i, j)$.

As such, we resort to an efficient alternative algorithm approximating effective resistance (Vos, 2016) for $\rho_{ij}$. In our research, we only need the appearance probabilities of all the edges to calculate the weight of the pairwise energies, so we only need to calculate the effective resistance between directly connected nodes instead of do the calculation between all the nodes. **Note**, the value $\text{Reff}(i, j)$ represents the probability of *sampling* edge $(i, j)$ when reaching node $i$ or node $j$ in Wilson's algorithm. In this sense, it is unnecessary to directly reweight the pairwise terms $\theta_{ij}$. More details are in **Appendix E**.

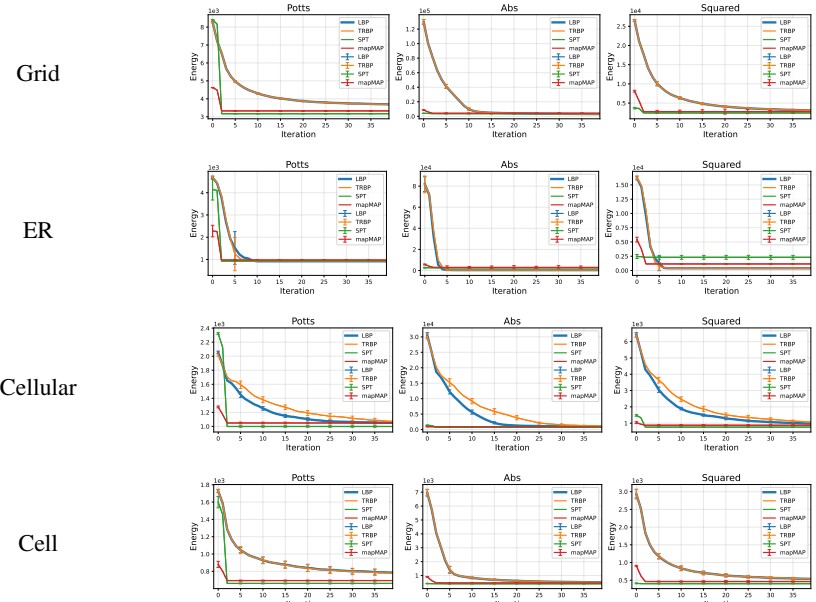

Figure 2: Results on Grid, Erdős-Rényi (ER), Cellular, and Cell graph types (rows) with Potts, Absolute difference (Abs), and Squared difference energies. Each plot shows energy $E(x|\bar{\theta})$ vs. iterations. Blue: LBP, orange: TRBP, green: SPT. The error bars show the standard deviation of each algorithm. Note: LBP and TRBP curves sometimes overlap due to minimal differences.

## 5 EXPERIMENTS

We present a comprehensive investigation into the performance of the variational inference method Mean-field and three belief-based inference algorithms: the Loopy Belief Propagation algorithm (LBP), the Tree-reweighted Belief Propagation algorithm (TRBP), and a novel sampling algorithm proposed in this study, referred to as **SPT**. The experiments aim to validate the arguments presented in the previous sections and establish the superiority of our proposed algorithm over LBP and TRBP. Performance evaluations are conducted on synthetic problems with diverse graph structures, encompassing energy functions such as the square function $\theta_{ij} = \alpha(x_i - x_j)^2$, absolute difference function $\theta_{ij} = \alpha|x_i - x_j|$ and Potts function $\theta_{ij} = \alpha(x_i = x_j) + \beta$, where $\alpha$ and $\beta$ are constant penalty terms. In this section, we use $\alpha = 1$ and $\beta = 0.1$.

Experiments involve grid-based and cell-based structures, as well as Erdős-Rényi (ER) random graphs. The LBP algorithm and TRBP algorithm are from Kappes et al. (2015); Andres et al. (2012); Maxwell Forbes (2017). TRBP here is the refined version of TRW-based algorithms. The mean-field algorithm we used is from Wang et al. (2022) and the mapMAP solver is from Thuerck et al. (2016). Furthermore, the applicability of our algorithm on real-world MRF inference datasets is demonstrated.

### 5.1 SYNTHETIC PROBLEMS

We first conduct experiments on synthetic problem instances. To test the applicability of the proposed method in a wide spectrum of settings, we test four types of graph topologies: Grid, Erdős-Rényi (ER), Cellular, and Cell. There are illustrative figures in **Appendix O.1** visualizing typical layouts of cell graphs (Fig. 7(a)) and cellular graphs (Fig. 7(b)). All the experimental results are in Fig. 2. Concrete settings are as follows.

**Grid.** We first test these algorithms on $70 \times 70$ grids with 4900 nodes, and each random node has 5 possible discrete labels to select. In addition, each label is attached to an explicit energy generated randomly and the energy $e_l \in (0, 1)$. For the setting of LBP and TRBP, we set the maximum round of iterations to 40 rounds. The damping factor of both of them is 0.8 since it is the best configuration as we could see from the results of the experiments. For our SPT, the maximum round of iterations is set to 20 and the damping factor is 0. We run the experiment for 10 different test cases and the final result is the average result over these cases.

**Erdős-Rényi.** Random MRF instances under Erdős-Rényi setting comprise 2,500 nodes, with an average degree of 12. The test cases are configured with different energy functions, and all the unary energy terms are generated randomly. In order to show the inference ability clearly, we attach 10 possible labels for each node

Table 1: Results on the UAI dataset. Numbers are the obtained energy values. Best in bold.

| GRAPH | #NODES/#EDGES | MEAN-FIELD | LBP | TRBP | mapMAP | SPT |
|---|---|---|---|---|---|---|
| PROTEINFOLDING_11 | 400/7160 | 1.0414E+08 | **7.11675E+07** | 7.54224E+07 | 8.54620E+07 | 7.49639E+07 |
| PROTEINFOLDING_12 | 250/1848 | 5852.2148 | **0.949** | **0.949** | **0.949** | **0.949** |
| GRIDS_19 | 1600/3200 | 10613.0645 | 7053.72 | **6056.45** | 7523.18 | 6275.56 |
| GRIDS_21 | 1600/3200 | 3.3756E+08 | 2.5577E+08 | **2.56852E+08** | 3.33740E+08 | 3.22289E+08 |
| GRIDS_24 | 1600/3120 | 3.8874E+08 | 3.08505E+08 | **3.01668E+08** | 3.53902E+08 | 3.09172E+08 |
| GRIDS_25 | 1600/3120 | 10487.6113 | 5476.03 | **5441.52** | 7635.35 | 5931.11 |
| GRIDS_26 | 400/800 | 818490.187 | 792987 | **556629** | 794293 | 759351.5 |
| GRIDS_27 | 1600/3120 | 3.892E+06 | **2.06277E+06** | 2.14945E+06 | 3.54505E+06 | 3.41471E+06 |
| GRIDS_30 | 400/760 | 1044407.0625 | **567339** | 663905 | 817746 | 809344 |
| SEGMENTATION_11 | 228/624 | 401.231 | 346.647 | 348.097 | 286.022 | **200.866** |
| SEGMENTATION_12 | 231/625 | 805.6727 | 735.259 | 735.259 | 724.025 | **611.347** |
| SEGMENTATION_13 | 225/607 | 785.7468 | 726.144 | 726.144 | 7689.666 | **596.606** |
| SEGMENTATION_14 | 231/632 | 803.2781 | 742.241 | 742.241 | 684.055 | **629.95** |
| SEGMENTATION_15 | 229/622 | 314.47 | 362.61 | 362.61 | 277.087 | **191.924** |
| SEGMENTATION_16 | 228/610 | 776.0317 | 720.009 | 720.009 | 647.788 | **578.114** |
| SEGMENTATION_17 | 225/612 | 516.1681 | 392.83 | 370.852 | 296.107 | **187.432** |
| SEGMENTATION_18 | 235/647 | 820.7836 | 782.282 | 782.282 | 662.122 | **595.11** |
| SEGMENTATION_19 | 228/624 | 798.2877 | 742.197 | 742.197 | 660.399 | **624.649** |
| SEGMENTATION_20 | 232/635 | 578.8339 | 373.14 | 371.839 | 287.187 | **196.558** |

in this part. The damping factors are set to 0 in this part of experiment. The maximum iterations, including our SPT algorithm are all set to 40. The number of trees used by SPT for the tests is 20.

**Cellular & Cell.** In addition to these two most common type of networks. We then run the tests on cellular and cell networks which are widely applied in communication networks. Each of the cellular networks has 4,998 nodes and 7,398 edges, the average degree is about 2.96. Each of the cell networks has 4,900 nodes and 14,421 edges, the average degree is about 5.89. The other settings are as same as we used for the experiments on Erdős-Rényi graphs.

**Results & Analysis.** The results are displayed in Fig. 2. These charts show our algorithm effectively handles various energy functions on graphs with specific local structures and randomly generated graphs. For grid, cellular, and cell graphs (all locally and sparsely connected), our algorithm outperforms both LBP, TRBP and mapMAP in final energy and iteration efficiency. These structures commonly appear in wireless communication and traffic control industries, where strategically placed interconnected devices create potential interference patterns similar to our experimental setup. Our algorithm performs consistently well on completely randomly generated graphs, even when graph density is approximately twice that of cellular networks. In addition, the error bars indicate that these algorithms remain highly stable after convergence, with our SPT algorithm exhibiting particularly robust performance. However, it slightly underperforms compared to LBP and TRBP when using absolute difference as the energy function. This occurs because when different choices have similar costs, the algorithm may converge to sub-optimal positions.

## 5.2 UAI INFERENCE COMPETITION

We extend evaluation to factor graphs from the 2022 UAI Inference Competition's MAP tasks[1]. Table 1 compares Mean-field, LBP, TRBP, mapMAP, and our SPT method, reporting final energy values. Algorithm settings match those used on synthetic problems, except SPT uses 20 trees for estimation in this section. Overall, SPT achieves competitive performance comparable to LBP and TRBP, with significant improvements on various segmentation tasks. On segmentation instances, SPT outperforms all other methods. These instances have graph structures similar to those in the synthetic experiments and share the same Potts model characteristics. For ProteinFolding_11 and Grid cases, our algorithm's performance matches LBP and TRBP, consistent with findings from Grid graph experiments in the synthetic problem set. Graph structures for ProteinFolding_11, ProteinFolding_12, Segmentation_14 and Grids_30 instances are shown in Appendix O.2. We display only one Segmentation class instance since they all share similar graph structure.

## 5.3 REAL-WORLD PCI PROBLEM OF 5G NETWORKS

PCI (Physical Cell Identity) uniquely identifies cells in LTE and 5G networks. We evaluate using internal PCI data transformed into pairwise MRFs for MAP inference. Appendix K details the MIP formulation and transformation process. We evaluate our algorithm on four internal *real industry-level PCI instances* compared to LBP and TRBP, using final energy (corresponding to the objective function in equation 52) as our comparison standard. Results appear in Table 2 with instance topologies in Appendix O.3. When MAP estimation is formulated as integer programming, problem size increases significantly, as shown in the table. Both LBP and SPT yield identical results, but our algorithm achieves superior performance compare to all the baselines, with

---

[1]https://www.auai.org/uai2022/uai2022_competition

Table 2: Results on real PCI instances. Numbers are obtained energy values. Best in bold.

| GRAPH | #VAR/#CON | #NODES/#EDGES | LBP | TRBP | MAPMAP | SPT |
|---|---|---|---|---|---|---|
| PCI_INSTANCE_1 | 955/2496 | 30/165 | 3.72662E+08 | 3.72662E+08 | 101259.1 | **84382.6** |
| PCI_INSTANCE_2 | 1588/4409 | 40/311 | 3.72704E+08 | 3.72704E+08 | 214875 | **186848** |
| PCI_INSTANCE_3 | 17684/52673 | 80/1522 | 0.303468 | 0.303468 | 0.299465 | **0.295245** |
| PCI_INSTANCE_4 | 65713/193287 | 286/10565 | 0.751074 | 0.751074 | 0.725626 | **0.552074** |

Table 3: Inference time results on the UAI dataset.

| GRAPH | #NODES/#EDGES | Mean-field | LBP | TRBP | mapMAP | SPT | SPT $\rho_{ij}$ Computation Time |
|---|---|---|---|---|---|---|---|
| ProteinFolding_11 | 400/7160 | 0.0343s | 0.8802s | 0.9671s | 0.0400s | 13.2087s | 1.6689s |
| ProteinFolding_12 | 250/1848 | 2.2345s | 13.4593s | 15.4034s | 0.1060s | 15.9054s | 0.4481s |
| Grids_19 | 1600/3200 | 0.1503s | 3.3206s | 3.6675s | 0.0930s | 112.9450s | 89.1308s |
| Grids_21 | 1600/3200 | 0.1598s | 3.3087s | 3.6697s | 0.1410s | 137.6740s | 89.1060s |
| Grids_24 | 1600/3120 | 0.1423s | 3.2266s | 3.5659s | 0.0460s | 142.9210s | 89.0508s |
| Grids_25 | 1600/3120 | 0.1337s | 3.0990s | 3.4754s | 0.0710s | 115.1980s | 90.8171s |
| Grids_26 | 400/800 | 0.0317s | 0.8243s | 0.9149s | 0.0480s | 12.3774s | 1.6195s |
| Grids_27 | 1600/3120 | 0.1375s | 3.2234s | 3.5731s | 0.1050s | 132.4540s | 89.1219s |
| Grids_30 | 400/760 | 0.0313s | 0.7504s | 0.8320s | 0.0400s | 13.4867s | 1.6566s |
| Segmentation_11 | 228/624 | 0.2086s | 24.8519s | 29.6790s | 0.0240s | 12.2192s | 0.3664s |
| Segmentation_12 | 231/625 | 0.0212s | 0.8221s | 0.8991s | 0.0180s | 3.7285s | 0.3779s |
| Segmentation_13 | 225/607 | 0.0208s | 0.6320s | 0.7084s | 0.0270s | 3.0541s | 0.3475s |
| Segmentation_14 | 231/632 | 0.2290s | 0.7520s | 0.8248s | 0.0210s | 3.6365s | 0.3756s |
| Segmentation_15 | 229/622 | 0.1961s | 24.7640s | 29.4172s | 0.0230s | 9.8776s | 0.3650s |
| Segmentation_16 | 228/610 | 0.2110s | 0.8111s | 0.8860s | 0.0260s | 3.5368s | 0.3630s |
| Segmentation_17 | 225/612 | 0.1895s | 25.8389s | 30.3104s | 0.0220s | 9.6708s | 0.3482s |
| Segmentation_18 | 235/647 | 0.0235s | 0.7459s | 0.8250s | 0.0310s | 3.6849s | 0.3962s |
| Segmentation_19 | 228/624 | 0.2210s | 0.7255s | 0.8071s | 0.0230s | 3.3923s | 0.3606s |
| Segmentation_20 | 232/635 | 0.3630s | 26.8320s | 31.4758s | 0.0230s | 10.7515s | 0.3682s |

substantial energy reductions for PCI_INSTANCE_1 and PCI_INSTANCE_2. The inference time are shown in Table. 5 in Appendix. J.

## 5.4 MORE ANALYSIS

To investigate the impact of various factors, we examine the following aspects: the number of trees used, the number of iterations, stopping criteria, other sub-graph structures, and impact of sparsity. "The main part of the result figures in this section is displayed in Fig. 3.

**Number of trees**. We evaluated our algorithm's performance on the PCI_INSTANCE_2 model using 1, 5, 20, 80, 160 spanning trees. After 10 iterations per configuration, the final energy values revealed a V-shaped curve, with optimal performance at 5 spanning trees (Fig. 3a). This suggests that for complex graphs, approximately 5-10 spanning trees provide sufficient structure representation without redundant information. As demonstrated in Appendix H, for trivial MRFs with known optimal energies, increasing the number of trees can further minimize the optimality gap.

**Number of iterations & Stopping criterion**. Fig. 3b shows energy curves for our algorithm on the PCI_INSTANCE_1 dataset using 20 spanning trees over 40 iterations. The optimal configuration is achieved at iteration 19 and persists until the end. Notably, the provisional best configuration remains stable during several periods within the first 20 iterations. Common early stopping criteria based on monitoring consecutive iterations without improvement (marked by red, green, and yellow dotted lines at thresholds of 3, 5, and 10 iterations) would fail to capture the best solution found at iteration 19. Given our algorithm's efficiency, we recommend running it for the maximum number of iterations in practice.

**Inference Time**. In Table 3, we show both overall inference time and edge selection probability calculation time. For complex problems like "Segmentation_11", all methods require more computation time, but SPT demonstrates more stable performance and faster convergence. Our analysis identifies calculating edge selection probabilities as the main computational bottleneck, especially for larger graphs. Developing more efficient probability estimation methods is an important future research direction. The algorithm could be accelerated through parallelization, with details available in Appendix M

**Other sub-graph structures**. We evaluated several structural alternatives: chains, random trees (covering 75% of nodes), and random walks (1600 steps for greater node coverage). Testing details are in Appendix. G. Tests used 1600-node grids with 10-16 possible states per node and square function pairwise energy without unary terms. Unlike our fixed-tree SPT approach, alternative structures were resampled each iteration since they don't guarantee full node coverage. Running 20 iterations with SPT using 10 trees, Fig. 3c shows SPT achieved superior results with significantly fewer iterations. Random trees and walks delivered acceptable results but required more iterations to converge. Comparison with the tree-coupling approach from Hamze & de Freitas (2004) appears in Appendix F.

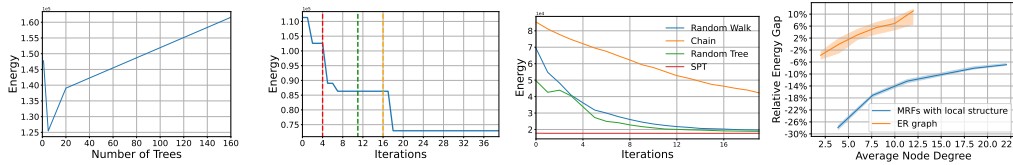

(a) Energy v.s. tree numbers  (b) Energy v.s. iterations  (c) Performance by different subgraphs  (d) SPT-LBP gap vs. sparsity

Figure 3: Part of the results for the analysis in Section. 5.4.

**Impact of sparsity**. From Fig.2, our algorithm outperforms LBP and TRBP on both grid and cell structures, with a larger performance gap on denser cell graphs. To evaluate effectiveness on locally and sparsely connected MRFs (Fig.8, Appendix O) and analyze local structure complexity impact, we tested LBP and SPT on graphs described in Appendix L. These graphs had average degrees of 3.8, 7.6, 11.3, 18.5, and 22.0. SPT used 10 trees, with 10-16 states per random variable and nodes linked to random observations. Pairwise energy used the Potts function. Performance comparison used the relative energy gap (equation 16), averaged over 10 trials.

$$\text{GAP} = (E_{SPT} - E_{LBP})/E_{LBP} \times 100\% \tag{16}$$

Fig. 3d shows that as local structures become denser, SPT increasingly outperforms LBP. This occurs because loop impacts become more significant, making it crucial to balance variable values within local graph structures—a strength of our algorithm. The shaded areas indicate modest gap variances.

Our algorithm also performs effectively on random graphs, especially when they're not highly dense. We conducted similar experiments on ER graphs with 900 nodes, varying average node degrees from 2 to 12 using squared energy functions (as performance with Potts functions is similar). Results in Fig. 3d (orange line) show our algorithm performs effectively on sparse graphs, even without specific graph topologies. The algorithm achieves satisfactory results on sparse graphs, but as average node degree increases, the performance gap between our algorithm and LBP increases linearly. Without local structure to leverage, SPT's performance is reduced.

## 6 CONCLUSION

MAP inference on MRFs remains a persistent challenge. We propose a novel, efficient approach for MAP estimation on locally and sparsely connected MRFs using spanning tree sampling. Our method works by solving MAP problems on individual sampled trees and merging these solutions into a final configuration. Experiments demonstrate superior performance against strong baselines across diverse settings, with particularly significant improvements on real-world PCI instances, indicating broad applicability potential.

## ACKNOWLEDGMENT

This work was supported by the National Key R&D Program of China under grant 2022YFA1003900.

## REPRODUCIBILITY STATEMENT

To ensure the reproducibility of our research findings, we have taken the following comprehensive measures:

Our implementation code has been made publicly available, with the repository link provided at the end of the abstract. This allows other researchers to directly access and utilize our algorithm for their own applications or verification purposes.

Detailed instructions for setting up and running our code are included in the repository documentation. These instructions cover all necessary steps from environment setup to execution of experiments, enabling smooth replication of our work.

In Section 5, we have explicitly documented all algorithm settings used in our experiments, including parameter configurations, optimization choices, and implementation details. These specifications are critical for reproducing our experimental results.

Access information for the UAI Inference Competition datasets used in our evaluation is clearly provided in Section 5.2. This includes source links and preprocessing steps applied to the data before running our experiments.

For synthetic experiments, we have thoroughly documented our instance generation methodology in Section 5.4, with additional technical details available in Appendix G. This documentation ensures that synthetic test cases can be precisely recreated.

We provide comprehensive instructions for transforming PCI problems from their original Mixed Integer Programming (MIP) format into the MRF format required by our algorithm. This transformation process is crucial for applying our method to real-world telecommunication network optimization problems.

Through these measures, we have ensured that all aspects of our research can be independently verified and built upon by the research community.

## ETHICS STATEMENT

During the preparation and submission of this paper, we have strictly adhered to the Code of Ethics in scientific research. We ensured proper citation of all relevant work, maintained integrity in our experimental procedures, reported results accurately without manipulation, and respected confidentiality of data sources where applicable. All authors have contributed substantially to this work and approved the final manuscript, with no conflicts of interest undisclosed.

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

## A    PROOF OF LEMMA 1

**Lemma 1.** *Given a uniform spanning tree distribution $\Omega(\mathcal{T})$ and the corresponding edge appearance probabilities $\{\rho_{ij}|\forall (i,j) \in \mathcal{E}\}$, when $|\mathcal{K}| = |\mathcal{T}|$, the approximation energy Eq. 9 and the original energy coincide.*

*Proof.* When $|\mathcal{K}| = |\mathcal{T}|$,

$$
\begin{aligned}
\tilde{E}(X) &= \sum_{i \in \mathcal{V}} \theta_i(x_i) + \sum_{(i,j) \in \mathcal{E}} |\{T_k | (i,j) \in T_k, \forall k \in \mathcal{K}\}| \frac{1}{\rho_{ij}} \theta_{ij}(x_i, x_j) \\
&= \sum_{i \in \mathcal{V}} \theta_i(x_i) + \sum_{(i,j) \in \mathcal{E}} \theta_{ij}(x_i, x_j),
\end{aligned}
\tag{17}
$$

which precisely is the original energy. $\qquad\square$

## B    PROOF OF LEMMA. 2

**Lemma 2.** *By applying the Monte Carlo Approximation, the expectation of the approximation is the original energy.*

$$
\mathbb{E}_{T_k \sim \Omega(\mathcal{T})} \left[ \sum_{i \in \mathcal{X}} \theta_i(x_i) + \frac{1}{|\mathcal{K}|} \sum_{k \in \mathcal{K}} \sum_{(i,j) \in T_k} w_{ij} \theta_{i,j}(x_i, x_j) \right] = \sum_{i \in \mathcal{X}} \theta_i(x_i) + \sum_{(i,j) \in \mathcal{E}} \theta_{ij}(x_i, x_j)
\tag{13}
$$

*Proof.* Since the unary term $\theta_i(x_i)$ is included in all spanning trees, we only need to prove the pairwise term in expectation is identical to the original pairwise energy term.

$$
\mathbb{E}_{T_k \sim \Omega(\mathcal{T})} \left[ \frac{1}{|\mathcal{K}|} \sum_{k=1}^{|\mathcal{K}|} \sum_{(i,j) \in \mathcal{T}_{\parallel}} w_{ij} \theta_{ij}(x_i, x_j) \right] = \frac{1}{|\mathcal{K}|} \sum_{k=1}^{|\mathcal{K}|} \mathbb{E}_{T_k \sim \Omega(\mathcal{T})} \left[ \sum_{(i,j) \in \mathcal{T}_k} w_{ij} \theta_{ij}(x_i, x_j) \right]
\tag{18}
$$

Since each tree is sampling uniformly and independently

$$
\frac{1}{|\mathcal{K}|} \sum_{k=1}^{|\mathcal{K}|} \mathbb{E}_{T_k \sim \Omega(\mathcal{T})} \left[ \sum_{(i,j) \in \mathcal{T}_k} w_{ij} \theta_{ij}(x_i, x_j) \right]
\tag{19}
$$

$$
= \frac{1}{|\mathcal{K}|} \sum_{k=1}^{|\mathcal{K}|} \mathbb{E}_T \left[ \sum_{(i,j) \in \mathcal{T}_k} w_{ij} \theta_{ij}(x_i, x_j) \right]
\tag{20}
$$

$$
= \mathbb{E}_T \left[ \sum_{(i,j) \in \mathcal{T}_k} w_{ij} \theta_{ij}(x_i, x_j) \right]
\tag{21}
$$

Here we use $\mathbb{E}_T$ to denote $\mathbb{E}_{T \sim \text{uniform}(T)}$.

Introduce indicator variables:

$$
I_{(i,j)}^{(k)} = \begin{cases} 1 & \text{if edge } (i,j) \text{ is in spanning tree } T_k \\ 0 & \text{otherwise} \end{cases}
\tag{22}
$$

We have $\mathbb{E}[I_{(i,j)}^{(k)}] = \rho_{ij}$.

Then,

$$\mathbb{E}_T \left[ \sum_{(i,j) \in \mathcal{T}_k} w_{ij} \theta_{ij}(x_i, x_j) \right] \tag{23}$$

$$= \mathbb{E}_T \left[ \sum_{(i,j) \in \mathcal{E}} w_{ij} I_{(i,j)}^T \theta_{ij}(x_i, x_j) \right] \tag{24}$$

$$= \sum_{(i,j) \in \mathcal{E}} w_{ij} \mathbb{E}_T [I_{(i,j)}^T] \theta_{ij}(x_i, x_j) \tag{25}$$

$$= \sum_{(i,j) \in \mathcal{E}} \frac{1}{\rho_{ij}} \rho_{ij} \theta_{ij}(x_i, x_j) \tag{26}$$

$$= \sum_{(i,j) \in \mathcal{E}} \theta_{ij}(x_i, x_j) \tag{27}$$

Therefore, you could see that the expectation of the approximation is the original energy function.

$\square$

## C    PROOF OF THEOREM 1

**Theorem 1.** *Given spanning tree distribution $\Omega(\mathcal{T})$ and the corresponding edge appearance probability $\{\rho_{ij} | \forall (i,j) \in \mathcal{E}\}$, the following error bound of the approximation energy Eq. equation 9 holds with probability at least $1 - \delta$.*

$$|E(X) - \tilde{E}(X)| \leq \sqrt{\frac{1}{|\mathcal{K}|} \sum_{(i,j) \in \mathcal{E}} \theta_{ij}^2(x_i, x_j) (\frac{1 - \rho_{ij}}{\rho_{ij}})} \frac{1}{\sqrt{\delta}}. \tag{14}$$

*When*

$$|\mathcal{K}| \geq \frac{1}{\delta \eta^2} \sum_{(i,j) \in \mathcal{E}} \theta_{ij}^2(x_i, x_j) (\frac{1 - \rho_{ij}}{\rho_{ij}}), \tag{15}$$

*we have $P(|E(X) - \tilde{E}(X)| \geq \eta) \leq \delta$.*

*Proof.* Given

$$E(X) = \sum_{i \in \mathcal{X}} \theta_i(x_i) + \sum_{(i,j) \in \mathcal{E}} \theta_{i,j}(x_i, x_j), \tag{28}$$

$$\tilde{E}(X) = \sum_{i \in \mathcal{X}} \theta_i(x_i) + \frac{1}{|\mathcal{K}|} \sum_{k \in \mathcal{K}} \sum_{(i,j) \in \mathcal{E}} w_{ij} \theta_{i,j}(x_i, x_j), \tag{29}$$

We have

$$\Delta E(X) = E(X) - \tilde{E}(X) \tag{30}$$

$$= (\frac{1}{|\mathcal{K}|} \sum_{k \in \mathcal{K}} \sum_{(i,j) \in \mathcal{E}} w_{ij} \theta_{i,j}(x_i, x_j)) - \sum_{(i,j)} \theta_{i,j}(x_i, x_j). \tag{31}$$

Introduce indicator variables:

$$I_{(i,j)}^{(k)} = \begin{cases} 1 & \text{if edge } (i,j) \text{ is in spanning tree } T_k \\ 0 & \text{otherwise} \end{cases} \tag{32}$$

Then the error becomes:

$$\Delta E(X) = \sum_{(i,j) \in \mathcal{E}} ((\frac{1}{|\mathcal{K}|} I_{(i,j)}^{(k)} w_{ij} \theta_{ij}(x_i, x_j)) - \theta_{ij}(x_i, x_j)). \tag{33}$$

Now define the Per-Edge error $\Delta E_{ij}$. Then

$$\Delta E(X) = \sum_{(i,j) \in \mathcal{E}} \Delta E_{ij}. \tag{34}$$

Define random variable $Z_{ij}^{(k)} = (I_{(i,j)}^{(k)} w_{ij} - 1)\theta_{ij}(x_i, x_j)$. Then,

$$\Delta E_{ij} = \frac{1}{|\mathcal{K}|} \sum_{k \in |\mathcal{K}|} \Delta Z_{ij}^{(k)}. \tag{35}$$

Since $\mathbb{E}[I_{ij}^{(k)}] = \rho_{ij}$

$$\mathbb{E}[Z_{ij}^{(k)}] = (\mathbb{E}[I_{ij}^{(k)} w_{ij}] - 1)\theta_{ij}(x_i, x_j) \tag{36}$$
$$= (\rho_{ij} w_{ij} - 1)\theta_{ij}(x_i, x_j) \tag{37}$$
$$= 0. \tag{38}$$

Then,

$$Var[Z_{ij}^{(k)}] = (w_{ij}\theta_{ij}(x_i, x_j))^2 \rho_{ij}(1 - \rho_{ij}) \tag{39}$$

$$Var[Z_{ij}^{(k)}] = (\frac{1}{\rho_{ij}}\theta_{ij}(x_i, x_j))^2 \rho_{ij}(1 - \rho_{ij}) \tag{40}$$

$$Var[\Delta E_{ij}] = \frac{Var[Z_{ij}^{(k)}]}{|\mathcal{K}|} \tag{41}$$

$$Var[\Delta E(X)] = \sum_{(i,j) \in \mathcal{E}} Var[\Delta E_{ij}]. \tag{42}$$

By Chebyshev's Inequality,

$$P(|\Delta E(X)| \geq \eta) \leq \frac{Var[\Delta E(X)]}{\eta^2} \tag{43}$$

$$\leq \frac{1}{|\mathcal{K}|\eta^2} \sum_{(i,j) \in \mathcal{E}} \theta_{ij}^2(x_i, x_j) \frac{(1 - \rho_{ij})}{\rho_{ij}}. \tag{44}$$

To ensure that the probability of the error exceeding $\eta$, is less than or equal to $\delta$,

$$\frac{1}{|\mathcal{K}|\eta^2} \sum_{(i,j) \in \mathcal{E}} \theta_{ij}^2(x_i, x_j) \frac{(1 - \rho_{ij})}{\rho_{ij}} \leq \delta. \tag{45}$$

Solving for $|\mathcal{K}|$,

$$|\mathcal{K}| \geq \frac{1}{\delta\eta^2} \sum_{(i,j) \in \mathcal{E}} \theta_{ij}^2(x_i, x_j) \frac{(1 - \rho_{ij})}{\rho_{ij}}. \tag{46}$$

Then, the error bound for the adjusted energy approximation E(X )is,

$$|\Delta E(X)| \leq \sqrt{\frac{1}{|\mathcal{K}|} \sum_{(i,j) \in \mathcal{E}} \theta_{ij}^2(x_i, x_j) \frac{(1 - \rho_{ij})}{\rho_{ij}}} \cdot \frac{1}{\sqrt{\delta}}. \tag{47}$$

This bound holds with probability at least $1 - \delta$. $\qquad\square$

## D   COMPLEXITY ANALYSIS

The algorithm consists of three parts: calculating the effective resistance, sampling spanning trees, and applying belief propagation on the spanning trees. Generally, the time complexity of computing the effective resistance is $O(MN^3)$. For sparse graphs, the factor $M$ can be considered a negligible coefficient. However, in scenarios involving large and complex graphs, the computational complexity of calculating the probability matrix becomes dominant relative to other operations. This computational burden represents the primary limitation for the broader application of our proposed method.

For the second step, the time complexity depends on the chosen spanning algorithm. In our approach, we adopt the method proposed in (Wilson, 1996), which has a runtime of $O(N^3)$. As discussed in Section. 4.2, Depth-First Search (DFS) was proposed as a potential approach to enhance computational efficiency at this stage. However, empirical results indicate that the performance difference between these two sampling methods is relatively modest in practice. This can be attributed to the availability of optimization techniques and the fundamental similarity between random walk-based sampling and DFS-based approaches.

The time complexity of the belief propagation is determined by the size of the spanning tree, which remains fixed when the graph is constant. Given $|\mathcal{E}_{\text{tree}}| = N - 1$, the time complexity becomes $O(N)$. It is worth noting that our algorithm converges rapidly, typically concluding within fewer than 10 iterations in most cases. Assume the average number of dependencies of each node is $b$, the time complexity of gibbs sampler would be $O(iter \times Nk)$. This efficiency allows us to disregard the term $iter$ without significantly affecting the complexity of the algorithm. As a result, the overall time complexity of the algorithm is $O(MN^3 + N)$.

## E    EFFECTIVE RESISTANCE CALCULATION

The calculation of resistance distance generally follows the procedure outlined in Theorem 2. The main contributor to the time complexity of this part is the computation of the Moore-Penrose inverse, as we need to use the effective resistance on all the edges of the graph.

**Theorem 2** (**Theorem 2.7 in** (Vos, 2016)). *The effective resistance between a pair of vertices $(i, j)$ is defined as $Reff_{i,j} := \Gamma_{i,i} + \Gamma_{j,j} - \Gamma_{i,j} - \Gamma_{j,i}$, where $\Gamma = (L + \frac{1}{|\mathcal{V}|}\Phi)^\dagger$, with $^\dagger$ denotes the Moore-Penrose inverse, $L$ the Laplacian matrix of $\mathcal{G}$, $|\mathcal{V}|$ is the number of vertices in $\mathcal{G}$, and $\Phi$ is the $|\mathcal{V}| \times |\mathcal{V}|$ matrix contain all $1$s. $\Gamma_{i,j}$ is the $(i, j)$ entry of the Moore-Penrose inverse of the Laplacian matrix.*

## F    COMPARISON TO TREE SAMPLING ALGORITHM

As previously discussed, the Tree Sampling algorithm proposed by Hamze & de Freitas (2004) is applicable only to graphs with specific structures that can be divided into two cycle-free parts. Under these conditions, the algorithm focuses on solving the MAP estimation problems on grid graphs with observation nodes. The MAP estimate is given by:

$$\min_X E(X) = \min_{\{x_i\}, \forall i \in \mathcal{V}} \left\{ \sum_{i \in \mathcal{V}} \theta_i(x_i, y_i) + \sum_{(i,j) \in \mathcal{E}} \theta_{ij}(x_i, x_j) \right\}. \tag{48}$$

here $y_i$ is the observation of the node $i$ which is deterministic value and $y_i, i \in \mathcal{V}$ has the same value range as the $x_i, i \in \mathcal{V}$.

To comprehensively evaluate the performance of our Spanning Tree algorithm and the Tree Sampling algorithm, we conducted experiments using grid graphs of varying sizes: 10×10 grids with 100 nodes, 20×20 grids with 400 nodes, and 30×30 grids with 900 nodes. Each node in these grids was assigned one of 10 possible labels. The pairwise energy and the energy between variables and their observations were defined using a squared label difference function. Observations were generated randomly for each instances.

For each grid size, we repeated the experiment across 10 different random instances to ensure robustness and reliability of the results. Each of these two algorithms run for 20 iterations. The number of spanning trees used by SPT is 10. As illustrated in Fig. 4, our algorithm consistently outperformed the Tree Sampling algorithm across all grid sizes. Importantly, the performance gap between the two algorithms remained consistent as the grid size increased.

## G    GENERATION OF DIFFERENT SUB-GRAPH STRUCTURES

In our previous discussions, we proposed using cycle-free structures to decompose the original graph, allowing us to leverage problems that can be solved exactly as approximations for the original problem. In addition to the advantages we outlined earlier, the spanning trees have demonstrated an extraordinary ability to provide more accurate approximations. We have also explored other structural alternatives, including chains, random trees (not necessarily spanning trees), and **random structures** generated using random walks. To obtain the **chains**, we used the following procedure: we would start at a randomly selected node and then perform a depth-first search to traverse the graph and collect the subsequent nodes. When sampling **random trees**, we would stop the sampling procedure once the number of nodes in the generated tree reached our target threshold. In this process, we allowed an agent to walk along the edges of the graph for a certain number of steps. After the agent stopped, the nodes it had visited and the edges it had traversed were extracted to form the sub-graph.

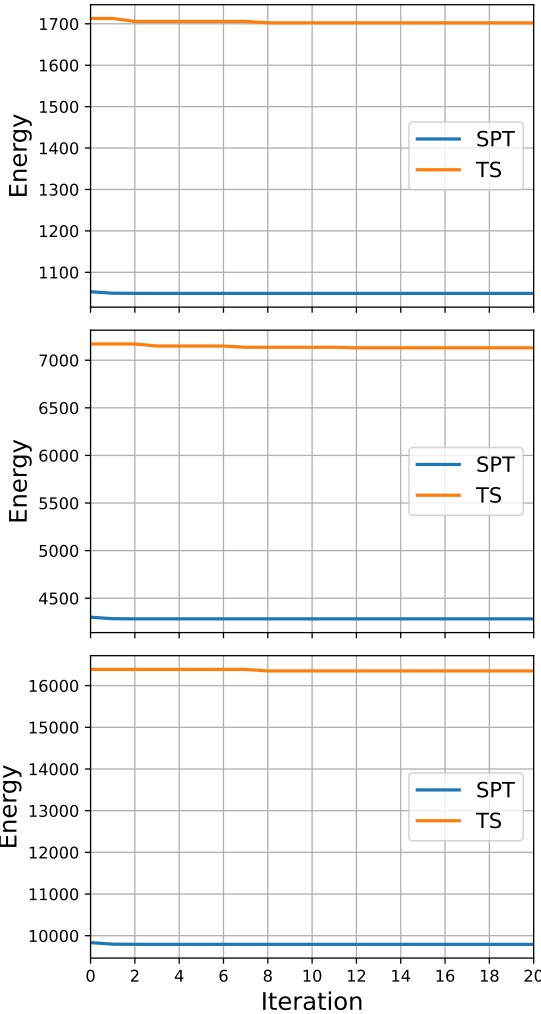

Figure 4: Results of SPT and TS on synthetic grid problems with squared label difference. The horizontal and vertical axes in the figure correspond to #number of iterations and average value of energy $E(x|\tilde{\theta}, y)$, respectively. From top to bottom, the grid sizes are 10×10, 20×20, 30×30.

## H    ERROR GAP ON TRIVIAL MRFS

We conducted additional experiments on MRFs with 64 nodes and increased the number of trees used to 100. We use the optimality gap as the evaluation standard. The results are listed in the Table 4. You can see that as more trees are sampled, the inference results converge to a point with only occasional fluctuations, and the absolute fluctuation is 1. This could be caused by the problem structure where some variable values offer similar energies that are difficult to distinguish, which is a characteristic of BP-based methods. Since our method is also based on BP, it is inevitable that we encounter the same issue. The calculation of Optimality gap is shown in Eq. 49 which could be found in the user manual of Mosek (ApS, 2025).

$$\text{Optimality Gap} = |\tilde{E}(X) - E^*(X)| \tag{49}$$

## I    EXAMPLE OF ENERGY WEIGHT ADJUSTMENT

In this section we use a toy example to show the importance of weight adjustment when using uniform spanning trees to decompose the original problem. In Fig. 5, we show a graph with four nodes and four edges, and we show all of the three spanning trees of it. Except edge $(1, 3)$, the probabilities of the other edges appear in a

Table 4: SPT inference optimality gaps with different number of trees on trivial MRF instances. Numbers are obtained optimality gaps.

| GRAPH | #NODES/#EDGES | 10 TREES | 20 TREES | 30 TREES | 40 TREES | 50 TREES | 60 TREES | 70 TREES | 80 TREES | 90 TREES | 100 TREES | OPT ENERGY |
|---|---|---|---|---|---|---|---|---|---|---|---|---|
| INSTANCE_1 | 64/143 | 5 | 6 | 3 | 2 | 2 | 3 | 2 | 2 | 3 | 2 | 94 |
| INSTANCE_2 | 64/156 | 7 | 5 | 5 | 5 | 5 | 5 | 5 | 5 | 5 | 5 | 134 |
| INSTANCE_3 | 64/124 | 1 | 2 | 2 | 2 | 2 | 2 | 2 | 1 | 1 | 1 | 104 |
| INSTANCE_4 | 64/116 | 3 | 4 | 5 | 6 | 5 | 3 | 3 | 3 | 3 | 2 | 131 |
| INSTANCE_5 | 64/133 | 1 | 1 | 2 | 2 | 2 | 1 | 2 | 2 | 2 | 1 | 117 |

uniform spanning tree are all $\frac{2}{3}$, which means the weights of the pairwise energies on these edges are all $\frac{3}{2}$. The probability of each of the trees being sampled is $\frac{1}{3}$. In Fig. 5 we using color red to denote the edges that have probability of $\frac{1}{3}$ being selected and the color blue to denote the edge that appear in all the uniform spanning trees.

Without loss of generality, we could assume the unary energies are all zero. Then we define the pairwise energies as follows. Each random variable has 2 possible states $\{0, 1\}$.

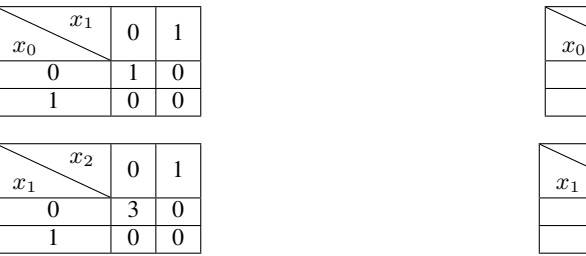

Now we calculate the energy when all the variables are at state 0. The original energy is $E(0,0,0,0) = 1+2+3+4 = 10$. If without weight, the energies on these trees would be $E_{T_1}(0,0,0,0) = 8$, $E_{T_2}(0,0,0,0) = 7$, $E_{T_3}(0,0,0,0) = 8$.

Then merging them together by

$$\tilde{E}(x) = \sum_{k \in \mathcal{K}} \rho_{T_k} \sum_{(i,j) \in T_k} \theta_{ij}(x_i, x_j) \tag{50}$$

$$\tag{51}$$

we can get the approximation value is $\frac{23}{3}$, which deviates from the original energy.

If we we adjust the pairwise energy using $1/\rho_{ij}$. the energies on these trees would be $E_{T_1}(0,0,0,0) = \frac{3}{2}(1+3)+4 = 10$, $E_{T_2}(0,0,0,0) = \frac{3}{2}(1+2)+4 = 8.5$, $E_{T_3}(0,0,0,0) = \frac{3}{2}(2+3)+4 = 11.5$. Then merging by Eq .equation 9, the approximation is $\frac{1}{3}(10+8.5+11.5) = 30$ which is exactly the original energy.

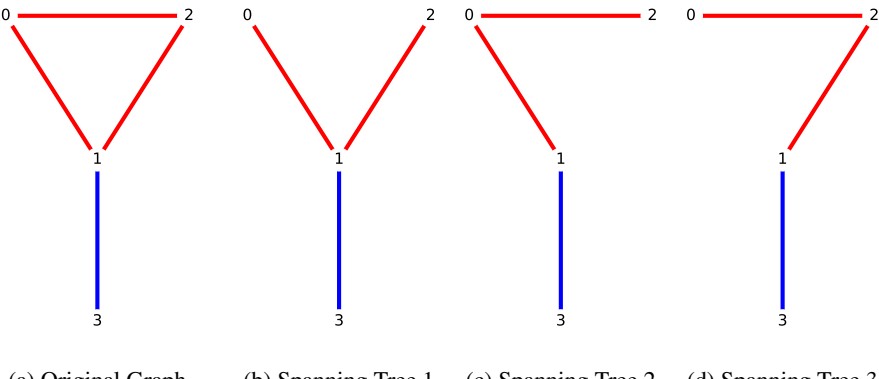

(a) Original Graph    (b) Spanning Tree 1    (c) Spanning Tree 2    (d) Spanning Tree 3

Figure 5: Schematic diagrams of spanning trees of a graph with cycle.

Table 5: Inference time on PCI instances.

| GRAPH | #VAR/#CON | #NODES/#EDGES | LBP | TRBP | SPT |
|---|---|---|---|---|---|
| PCI_INSTANCE_1 | 955/2496 | 30/165 | 0.359283s | 0.441703s | 0.405572s |
| PCI_INSTANCE_2 | 1588/4409 | 40/311 | 0.39226s | 0.652174s | 0.43437s |
| PCI_INSTANCE_3 | 17684/52673 | 80/1522 | 0.853267s | 0.91823s | 0.770843s |
| PCI_INSTANCE_4 | 65713/193287 | 286/10565 | 2.14694s | 6.211s | 3.609s |

## J    INFERENCE TIME ON PCI INSTANCES

In Table 5, we present the inference times of LBP, TRBP, and our SPT algorithm on the real PCI instances used in Section 5.3. Notably, despite SPT's $O(N^3)$ time complexity, its inference times on these instances fall between those of LBP and TRBP.

## K    TRANSFORMING MIP PROBLEMS OF PCI INTO MRF PROBLEMS

The Mixed Integer Programming(MIP) format of PCI problems is as follows:

$$\min_{z,L} \quad \sum_{(i,j)\in\mathcal{E}} a_{ij}L_{ij} \tag{52}$$

$$\text{s.t.} \quad z_{np} \in \{0,1\}, \quad \forall n \in N, p \in P \tag{53}$$

$$\sum_{p\in P} z_{np} = 1, \quad \forall n \in N. \tag{54}$$

$$\sum_{p\in M_{ih}} z_{n_i p} + \sum_{p\in M_{jh}} z_{n_j p} - 1 \leq L_{ij}, \forall (i,j) \in \mathcal{E}, \forall h \in \{0,1,2\}. \tag{55}$$

where $n$ is the index for devices, and $N$ is the set of these indices. $P$ stands for the possible states of each device. $M_{ih}$ stands for the possible states set for node $n_i$. $L_{ij}$ is the cost when given a certain choices of the states of device $i$ and device $j$, $a_{ij}$ is the coefficient of the cost in the objective function. There is an $(i,j) \in \mathcal{E}$ means there exists interference between these two devices. In the MIP formulation of the PCI problems, there are three types of constraints. Combining equation 53 and equation 54 together implies that each device must select one state, and only one state can be chosen at a given time. The constraint equation 55 indicates that interference occurs between two devices only if they choose specific states. The impact on the entire system is determined by the corresponding value of $L_{ij}$ and its coefficient. Since interference always exists, the objective is to minimize its degree.

To transform these problems into MRF problems, we can use equation 54 to represent nodes, where each equation 53 corresponds to the discrete states of a given node. Since only one state can be chosen at a time, the constraints equation 53 and equation 54 are naturally satisfied. By processing equation 55, we identify the edges and their associated energies. If we find $z_{n_i p}$ and $z_{n_j p}$ in the same constraint from equation 55, we can formulate an edge $(i,j)$. By selecting different values for $z_{n_i p}$ and $z_{n_j p}$, we can determine the minimum value of $L_{ij}$ that satisfies the constraint. The product of $L_{ij}$ and $a_{ij}$ represents the energy for the edge $(i,j)$ under the combination of these two states. When all the states of the nodes are fixed, the values of the edge costs become fixed as well. This implies that the objective function is the summation of all the edge energies. Since the PCI problems do not include unary terms, we will neglect them during the transformation process.

**Example**
The original problem is

$$\begin{aligned}
\min_{z,L} \quad & L_{1,2} + 2L_{2,3} \\
\text{s.t.} \quad & z_{np} \in \{0,1\}, & \forall n \in \{1,2,3\}, p \in \{1,2,3\} \\
& \sum_{p\in P} z_{np} = 1, & \forall n \in \{1,2,3\}. \\
& z_{11} + z_{21} - 1 \leq L_{1,2} \\
& z_{13} + z_{22} - 1 \leq L_{1,2} \\
& z_{12} + z_{23} - 1 \leq L_{1,2} \\
& z_{21} + z_{31} - 1 \leq L_{2,3} \\
& z_{22} + z_{32} - 1 \leq L_{2,3} \\
& z_{23} + z_{33} - 1 \leq L_{2,3}
\end{aligned} \tag{56}$$

Then the corresponding MRF problem is

$$\min \theta_{1,2}(x_1, x_2) + \theta_{2,3}(x_2, x_3) \tag{57}$$

the energy on edge $(x_1, x_2)$ and edge $(x_2, x_3)$ are as follows:

| $x_1$ \ $x_2$ | $z_{21}$ | $z_{22}$ | $z_{23}$ |
|---|---|---|---|
| $z_{11}$ | 1 | 0 | 0 |
| $z_{12}$ | 0 | 0 | 1 |
| $z_{13}$ | 0 | 1 | 0 |

| $x_2$ \ $x_3$ | $z_{31}$ | $z_{32}$ | $z_{33}$ |
|---|---|---|---|
| $z_{21}$ | 2 | 0 | 0 |
| $z_{22}$ | 0 | 2 | 0 |
| $z_{23}$ | 0 | 0 | 2 |

## L  GENERATION OF LOCALLY AND SPARSELY CONNECTED MRFS

We begin by uniformly placing the nodes in a 2-dimensional space. Next, we iteratively traverse all the nodes in an arbitrary order. For each node visited, we identify its k-nearest neighbors and create edges between them until the required average degree is achieved. Once all nodes have been visited, the graph construction is complete.

## M  PARALLELIZATION

When the size of the graph increases, our algorithms still function, but they become time-consuming. This is because the time complexity, as we have just analyzed, is highly dependent on both $N$ and $M$ (where $M$ is related to the number of states $n$ in the Markov chain). Evidently, acquiring comprehensive information about the entire graph would require more than one tree. Such a requirement could result in a notable slowdown of our algorithm due to the necessity of performing belief propagation on each tree. Nonetheless, the algorithm we have introduced can be seamlessly modified for parallel processing. Once the edge selection probabilities have been calculated, both the spanning tree sampling process and the subsequent belief propagation on each sampled tree can be performed independently and in parallel. This adaptability enables us to significantly improve efficiency.

## N  ITERATION WITH DIFFERENT BATCHES OF SPANNING TREES

Besides using fixed trees to perform the estimation, we could also generate another batch of trees to conduct the estimation in a new iteration, similar to what was done when using different subgraph structures for the estimation. However, this could be an expensive strategy to adopt. Note the analysis in **Appendix D**, where the time complexity of sampling a spanning tree is $O(MN^3 + N)$, which could incur a significant computational cost as the graph size increases.

To get a comprehensive understanding of whether we could gain more by spending more time on sampling trees and how the two strategies would perform with a similar time budget, we experimented on 40x40 grid graphs with 1600 nodes. Each random node has 10 to 16 possible states to choose from, the pairwise energy is determined by a square function, and the unary energy is set to zero. For the SPT method using fixed trees, the number of trees is set to 20. For the SPT with tree resampling, the number of trees is varied as {1, 5, 10, 15, 20}. All the SPT variants are run for 20 iterations.

As shown in Fig. 6, energy curves for each setting of the SPT methods reveal that both the SPT variants with 20 trees achieve the best and identical results. As the number of trees used per iteration increases, the gain on the final energy decreases, and the improvement rate diminishes quickly. However, the actual time cost of the SPT with fixed trees lies between the SPT with tree resampling using 1 tree and 5 trees and far less than re-sampling 20 trees at each iteration. This suggests that, although we could resample the trees during the iterative process, the additional time spent on this may not be worthwhile.

## O  INSTANCE TOPOLOGY

### O.1  SYNTHETIC PROBLEMS

Fig. 7 illustrates schematic diagrams of cellular ,cell graph and Erdős-Rényi graph structures. And in Fig. 8 illustrates two schematic diagrams of the locally and sparsely connected graphs we generated. Note they do not correspond to any testing instances.

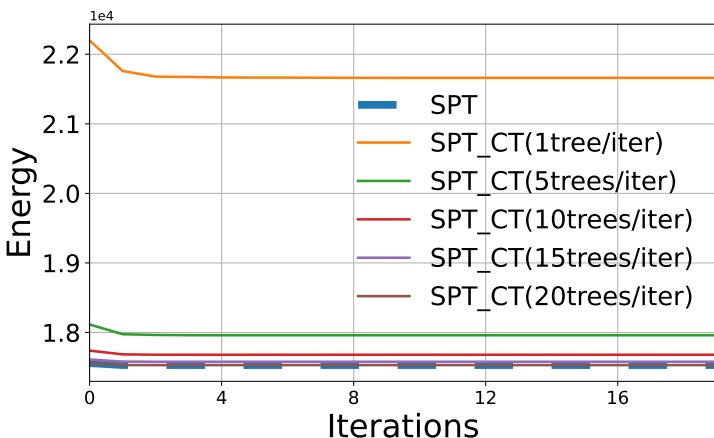

Figure 6: Energy curves of SPT using fixed trees or resample spanning trees at each iteration.

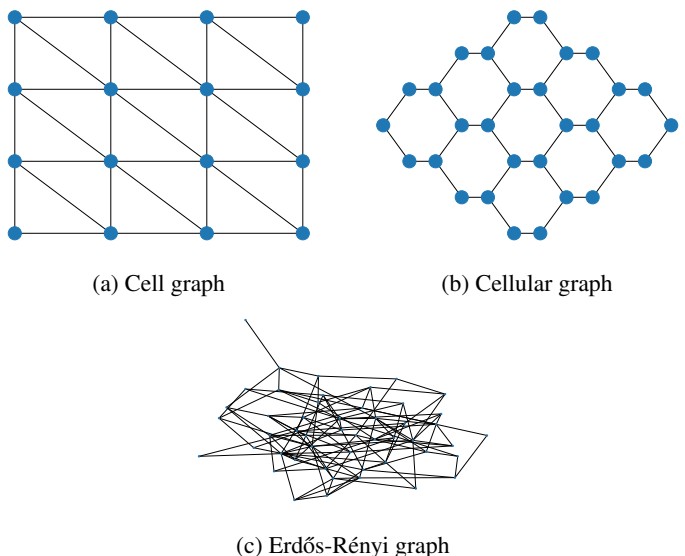

(a) Cell graph          (b) Cellular graph

(c) Erdős-Rényi graph

Figure 7: Schematic diagrams of cell graph, cellular graph and Erdős-Rényi graph.

## O.2    UAI INFERENCE COMPETITION

We visualize the topology of four instances from UAI competition in Fig. 9.

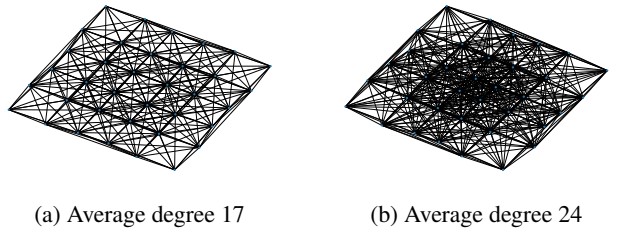

(a) Average degree 17          (b) Average degree 24

Figure 8: Schematic diagrams of locally and sparsely connected graphs.

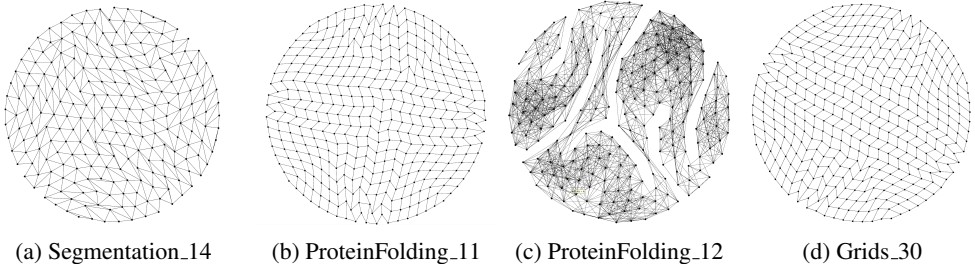

(a) Segmentation_14    (b) ProteinFolding_11    (c) ProteinFolding_12    (d) Grids_30

Figure 9: Schematic diagrams of some UAI inference competition instances. Zoom in for better view.

## O.3 PCI INSTANCES

All the topologies of the PCI instances are in Fig. 10.

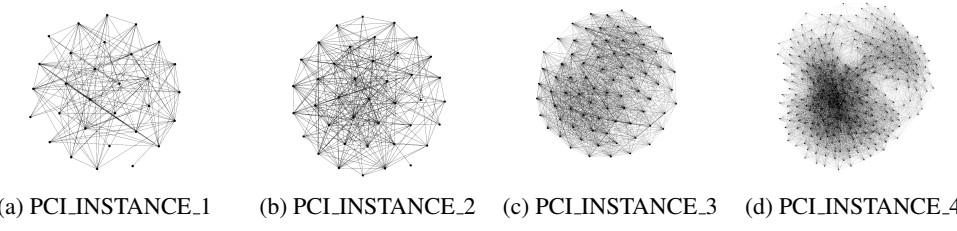

(a) PCI_INSTANCE_1    (b) PCI_INSTANCE_2    (c) PCI_INSTANCE_3    (d) PCI_INSTANCE_4

Figure 10: Schematic diagrams of PCI instances. Zoom in for better view.

## P LLM STATEMENT

In our research process, we utilized large language models primarily to support technical writing aspects rather than for generating research content. These tools were employed specifically for grammar checking, spell correction, improving sentence structure, and enhancing the overall readability of the manuscript. All scientific contributions, technical analyses, experimental designs, and conclusions presented in this paper are the original work of the authors, with language models serving only as writing assistance tools.

