# OpenReview forum: "From Fields to Random Trees"
_ICLR.cc/2026/Conference — ICLR 2026 Poster_

### Official Review · Reviewer_Vpu6 · 2025-10-27

**Soundness:** 3
**Presentation:** 3
**Contribution:** 3
**Rating:** 8
**Confidence:** 2

**Summary:**

This paper proposes a spanning-tree–based MAP inference algorithm (called SPT) for Markov Random Fields (MRFs) on locally and sparsely connected graphs, which is a regime common in power grids, communication networks, and transportation networks. The key idea is to sample uniform random spanning trees (RSTs) from the original graph; then run exact BP on each sampled tree (cycle-free so inference is tractable). Experiments on both synthetic an real datasets show the better performance of the proposal.

**Strengths:**

- The paper has clear motivation and introdution for the problem setup and the proposal.
- The complexity analysis and convergence guarantee is given and offer insights into the dependence on the problem parameters.
- The proposed algorithm is scalable thus supports inference for large graphs.
- The empirical validation on synthetic + UAI + real PCI datasets shows promising gains of the proposed approach.

**Weaknesses:**

See questions.

**Questions:**

- In Theorem 1, the error bound decreases as the number of spanning trees $\mathcal{K}$ increases. The proposed algorithm is taliored for sparse graphs. However, a sparse graph should have less spanning trees compared to a dense graph, which leads to a worse error bound. How do we understand this?
- In the experiment of Figure 1, where we compare energy against number of iteration, I wonder if this is a fair comparison. As these methods can be doing very different things and computation in each iteration step.
- Maybe a better comparison is to look at the wall clock time needed for each baseline to reach a certain energy accuracy.
- Why is the proposed algorithm inconsistent on ER graph?

---

> ### Author Response · Authors · 2025-11-21
>
> We appreciate your valuable feedback, which has significantly contributed to enhancing the quality of our work.
>
> > **Q1**
>
> The actual value of the error not only depends on the factor $\frac{1}{|\mathcal{K}|}$ but also the summation term $\sum \limits_{(i,j)\in \mathcal{E}} \theta_{ij}^2(x_i, x_j)(\frac{1-\rho_{ij}}{\rho_{ij}})$. In a sparse graph with fewer edges, there are fewer pairwise energy terms to sum, and the value $\rho_{ij}$ would be larger, which further reduces the summation. Therefore, it is difficult to characterize the error bound based solely on graph density.
>
> > **Q2/Q3**
>
> Your concern is actually very meaningful since clock time is one of the major concerns in practice. We plot energy versus iteration because our method and the baselines are all BP variants, so this view best illustrates their convergence behavior — and it shows that our algorithm converges faster. Clock-time results are reported in Table 2, and we have updated the inference times for the PCI instances in **Table.4 in Appendix.K in the updated pdf, where we show that the inference time of SPT lies between those of LBP and TRBP**; all reported runtimes are well below the UAI competition limit of 3600 s. **In many cases, our method is substantially faster: on nearly half of the segmentation_xx instances it is about three times faster than the two baselines.** Crucially, our method shows very stable runtime across instances, whereas LBP and TRBP can be up to 30× slower on graphs of comparable size.
>
> > **Q4**
>
> As stated in the paper, our SPT algorithm performs better on graphs that are locally and sparsely connected, such as grids and cellular structures. These graphs typically have well-defined node coordinates in low-dimensional space, which enable a natural notion of **locality** and correspond to various real-world physical networks (e.g., sensor arrays or cellular automata). In contrast, ER graphs are purely probabilistic random graphs with no inherent low-dimensional coordinate embedding, leading to structural uncertainty that manifests as high variability across different spanning trees. This variability makes it challenging for our approach—relying on a limited set of trees—to effectively capture the essential information of the original graph.

---

### Official Review · Reviewer_inkb · 2025-10-30

**Soundness:** 3
**Presentation:** 3
**Contribution:** 3
**Rating:** 6
**Confidence:** 2

**Summary:**

The paper explores sparsity and local connectedness in MAP estimation of Markov Random Fields (MRFs) proposing to combine multiple (spanning) subtrees learned exactly by sampling multiple spanning trees from the original graph and applying conventional belief propagation (BP). These exactly solved spanning subtrees are combined through an inverse weighting function accounting for the probability of sampling given edges to provide aggregated approximate solutions to the entire MRF. The approach is simple and straightforward to implement and appears to work well in practice as highlighted in the papers experimental section.

**Strengths:**

•	The approach is well related to the existing literature in the literature review.

•	The approach appears to work well in practice and is simple to implement.

•	The weighting scheme to correct for how edges in the spanning tree are sampled compared to uniform sampling appears as a simple, elegant, and valid practical approach to correct for biases by the sampling procedure.

Originality:
The approach combining multiple spanning subtrees appears new and original.

Quality:
The paper is generally clear and easy to follow. The experimentation considers comparison to a limited number of baselines which can be improved.

Clarity:
The paper can be improved in its writing - please also see the minor comments under Weaknesses, however, the developed methodology is clear.

Significance:
Minimizing pairwise MRFs are an extensively studied field with many contributions and approaches developed over the years. The paper here develops an interesting new approach which could warrant publication, but the significance of the results are unclear as no error bars are reported and the paper only compared to a limited number of alternatives.

**Weaknesses:**

The present procedure relies on estimating the spanning three exactly requiring O(N^3) which limits the approach to small graphs and subtrees. While scalable approaches are discussed it is unclear how well they perform in practice.

Whereas applications where the methodology is important is discussed in the motivation it is unclear how the solutions benefit practical applications. It would strengthen the paper to consider the solutions for at least one of the given problem domains highlighted in the motivation (introduction) and the practical implications.

The method is compared to very few competing methods, whereas the literature and approaches minimizing the energy of MRF including for the considered pairwise MRFs is vast as also highlighted in the rather old survey:

Wang, Chaohui, Nikos Komodakis, and Nikos Paragios. "Markov random field modeling, inference & learning in computer vision & image understanding: A survey." Computer Vision and Image Understanding 117.11 (2013): 1610-1627.

Where submodular random fields have also been considered for pairwise random fields such as the papers cited in the related works section:

H. Ishikawa, Exact Optimization for Markov Random Fields with Convex Priors, IEEE Transactions on Pattern Analysis and Machine Intelligence (TPAMI) 25 (10) (2003) 1333-1336.

D. Schlesinger, B. Flach, Transforming an Arbitrary Minsum Problem into a Binary One, Tech. Rep. TUD-FI06-01, Dresden University of Technology (2006)

Furthermore MRF estimation has been scaled using massively parallelized implementations as in:
https://download.mmag.hrz.tu-darmstadt.de/media/FB20/GCC/paper/Thuerck-2016-HPG.pdf

In this context, I find the experimentation to only include a very limited  number of comparisons to existing methods, i.e. Mean field, LBP, and TRBP whereas the literature is vast with many proposed methods.

Minor issues of the paper needing some proof-reading:
Problem equation 1 is known NP-hard in general -> The problem in equation 1 is known to be NP-hard in general

solve equation 1 can date back 80’s of the previous century -> solve equation 1 dates back  to the 80’s of the previous century
approach to infer on MRFs  -> approach to infer MRFs

Strange use of past tense of “could” in the Methods section which should be “can” throughout please check.

Since Acquire the true tree -> Since Acquiring the true tree

**Questions:**

Consider include scalable approaches based on approximate spanning tree estimation procedures and compare how the exact to such scalable approaches compare.

Please provide error bars in Figure 1 and the Tables across multiple runs. As the procedure is non-deterministic it will be good to see how much variability this induces on the results.

MRF estimation has been demonstrated to benefit from massively parallelization – how does the proposed approach compare to such parallelized implementations as in:

https://download.mmag.hrz.tu-darmstadt.de/media/FB20/GCC/paper/Thuerck-2016-HPG.pdf

Furthermore, how does the procedure compare to other methods covered in the related works section such as

H. Ishikawa, Exact Optimization for Markov Random Fields with Convex Priors, IEEE Transactions on Pattern Analysis and Machine Intelligence (TPAMI) 25 (10) (2003) 1333-1336.

D. Schlesinger, B. Flach, Transforming an Arbitrary Minsum Problem into a Binary One, Tech. Rep. TUD-FI06-01, Dresden University of Technology (2006)

In equation 12 the product does not produce as I understand it a valid conditional distribution \tilde{p}(x_i|X\{x_i}). Please clarify how samples are drawn from the distribution, is it simply renormalized by \tilde{p}(x_i|X\{x_i})/(\sum_{x_i} \tilde{p}(x_i|X\{x_i})) and shouldn’t "=" then be "\propto" (if it is not a normalized distribution)?

In summary, I consider this a borderline leaning accept paper but with room for improvements in particular in terms of establishing their approach to more alternatives by providing a wider comparison and include error bars for the assessments of results.

---

> ### Author Response · Authors · 2025-11-21
>
> Thank you for your insightful comments, which have greatly helped improve the quality of our work.
>
> > **W1/W2**
>
> Thank you for raising the important concern about actual efficiency and real-world applicability — these are primary evaluation targets for our method.
>
> In our experiments we strictly follow the UAI inference competition time limit (3600 s). Although our method sometimes uses more time than LBP and TRBP, the maximum inference time in the UAI dataset remains far below the limit (just over 100 s). **In many cases, our method is substantially faster: on nearly half of the segmentation_xx instances it is about three times faster than the two baselines.** Crucially, our method shows very stable runtime across instances, whereas LBP and TRBP can be up to 30× slower on graphs of comparable size (Table.2 in Section 5.2).
>
> For practical applications, we evaluate PCI problems in Section 5.3, a challenging 5G network device deployment task. The runtimes for these test cases are reported in **Table.4 in Appendix. K**, where we show that the inference time of SPT lies between those of LBP and TRBP, you can check the result in the updated pdf of the paper to give a clearer picture or you could just check the table below. Note that each deployment typically requires this computation only once, so total runtime is less critical than the quality of the final inference. Taken together, **these results indicate our algorithm is practical for complex, large-scale real-world problems.**
>
> | Graph |  #VAR/#CON | #NODES/#EDGES |    LBP |  TRBP |   SPT |
> | -------- | ---------- | ------------ | -------------- | --------------- | --------------- |
> | PCI\_INSTANCE\_1 |  955/2496 | 30/165 | 0.359283s   |   0.441703s    |    0.405572s  |
> | PCI\_INSTANCE\_2 |  1588/4409 | 40/311  |  0.39226s     |   0.652174s     |   0.43437s   |
> | PCI\_INSTANCE\_3 |  17684/52673 | 80/1522  | 0.853267s   |   0.91823s      |   0.770843s  |
> | PCI\_INSTANCE\_4 |  65713/193287 | 286/10565  | 2.14694s     |    6.211s    |  3.609s |
>
> > **W3/Q2**
>
> Thank you for suggesting additional baselines. In response, we have added the inference results of the mapMAP solver [Thürck et al., 2016] (from the paper you referenced) to Table 2 in Section 5.2. We were unable to locate publicly available code for the other two methods you mentioned. Given the tight timeline, we prioritized implementing and evaluating mapMAP, which provides a strong additional baseline. As shown in Table 2, mapMAP achieves inference times very close to our SPT algorithm on the Segmentation_xx instances, confirming the competitiveness of our approach while still demonstrating SPT's superior performance overall.
>
> > **W4.Writing issue**
>
> These suggestions are valuable. We have updated the pdf and changed the expression as you suggested.
>
> > **Q1.Error bar**
>
> We have updated the PDF to include **the revised Figure 1**. Figure 1 reports overall results for 10 test cases. For each case, every baseline and our algorithm is run once. As shown in the updated Figure 1, our algorithm exhibits stable performance across most test cases, with low standard deviation relative to the energy values. Moreover, it demonstrates greater stability compared to the other two baselines.
>
> > **Q2**
>
> Yes, you are right. The writing here needs to be fixed. Using "\propto" to is more appropriate since we didn't explicitly show the normalization step. We have fixed this in the updated pdf.

---

> ### Comment · Reviewer_inkb · 2025-11-24
>
> I thank the authors for their answers to my concerns and for including an additional baseline as well as produce examples of uncertainty in the revised figure 1.
>
> In light of the authors answers I maintain my assessment. It is unclear to me why error bars are still not reported in Table 2, but I understand the limited time makes comparison to procedures where code is not available infeasible and appreciate the updated inclusion of the work of [Thürck et al., 2016]. For completion could this approach also be added to Figure 1 and Table 1?

---

> ### Author Response · Authors · 2025-11-25
>
> Thank you for your understanding regarding the constrained time and heavy workload during the rebuttal phase. We have incorporated the inference results for mapMAP into both Table 1 and Figure 1 in the latest updated PDF. The baseline comparisons and uncertainty information you recommended have been invaluable in more effectively demonstrating the performance of our algorithm.

---

### Official Review · Reviewer_14zd · 2025-10-31

**Soundness:** 3
**Presentation:** 3
**Contribution:** 2
**Rating:** 6
**Confidence:** 3

**Summary:**

The paper studied the problem of finding the minimum energy state in a Markov random field (MRF). The problem is NP-hard in general because it can in-code the max-cut problem. The Belief Propagation (BP) is a famous algorithm for computing marginal distributions (inference) of MRFs. The algorithm can solve the inference problem exactly if the graph is a tree. The main contribution of this paper is a heuristic algorithm for finding the minimum energy state of a MRF.
- The algorithm generate some i.i.d. samples of uniform random spanning trees.
- For each tree (re-weight the edge according to the marginal of random spanning trees), run BP to compute marginals of every nodes.
- Merge the results of all trees by taking the product in (12) line 237.
- Given the marginals, using a greedy algorithm and a simple sampling (sample each marginal independently) algorithm to find a state. Update the current best answer.

The paper gives theoretical analysis on some step of the algorithm. The paper then did a lot of experiment on both synthetic data and real-world data.

To summarize, I think the theory part is of this paper is simple. The idea share some similarity with graph sparsification algorithm, which also use the random tree to sparsify the graph and change the edge weight (you may add some discussion). Some main theoretical results is the correctness of expectation and a concentration bound. However, it is interesting to see a simple algorithm works in many real application.

**Strengths:**

- The paper proposed to use random trees to approximate the MRF, which connects the concept of graph sparsification with the MAP (Maximum A Posteriori) inference problem in probabilistic graphical models. Intuitively, by constructing a collection of randomly sampled tree structures, one can run existing algorithms on trees while random trees preserves some information of the original MRF.

- The experimental evaluation on synthetic data and real-world data is comprehensive and well-structured. Detailed tables and visualizations are provided to illustrate the results, including comparisons with baseline methods. The findings shows that their algorithm can achieve a good performance in many situations.

**Weaknesses:**

- The MAP problem is trivial in trees because one can use a dynamic programming (the algorithm called max-product or min-sum algorithm). In this paper, the authors use BP to compute marginal distributions on trees and use marginal on trees to approximate marginal on MRFs.  The connection between computing marginal distributions and identifying the minimum energy configuration is not stated in the paper.  Even if this is a heuristic, it would be good to explain some intuition here. (See more detailed questions in the next section)

- Some statement in the theorems and lemmas and some proofs look confusing.  (See more detailed questions in the next section)

**Questions:**

Consider a Markov Random Field (MRF) defined on a general graph, inducing a Gibbs distribution $\mu$ over $\mathcal{X}^V$ (assume $\mathcal{X} =$ {0,1}). Although computing marginals is generally intractable, suppose we have access to an oracle that provides the marginal distribution for each node. For any $v \in V$, the oracle returns $p_v = \text{Pr}_{X \sim \mu}[X_v = 0]$. With this information, we can directly run the GibbsSampler and GreedySelector as described in Algorithm 1. The question is: does the algorithm find a good state with low energy?

Here is a simple example. Consider the energy function defined in (1), with $\mathcal{X}$ = {0,1} and $\theta_i(x_i) = 0$ for all $i$ and $x_i$. For the pairwise interaction term $\theta_{ij}$, define:
- $\theta_{ij}(x_i, x_j) = 1$ if $x_i \neq x_j$
- $\theta_{ij}(x_i, x_j) = 0$ if $x_i = x_j$

Clearly, the minimum energy configuration corresponds to the MAX-CUT of the graph, which is NP-hard. In this case, due to the symmetry between values $0$ and $1$, the marginal distribution for each node is uniform over {0,1}. This suggests that marginals alone may not help in finding the minimum energy state.

**Question:** In general, what is the relationship between computing marginals and finding the minimum energy state? Since the problem itself is NP-hard, we cannot expect the algorithm work for all cases. It is better to give some intuition and explanation on which situation this idea could provide a good solution.

---

**Lemma 1** relies on an unrealistic assumption. The set $\mathcal{K}$ is a randomly sampled set in the algorithm and may contain duplicate elements. The lemma assumes $\mathcal{K} = \mathcal{T}$, but this assumption is problematic for two reasons:
1. Let $N = |\mathcal{T}|$ be the number of spanning trees. Since $N$ can be exponential in $n$, i.e., $N = \exp(O(n))$, it is infeasible to generate that many samples.
2. Even if it were feasible, the condition $\mathcal{K} = \mathcal{T}$ implies no duplicates in $\mathcal{K}$ (otherwise, equation (17) in the proof of Lemma 1 fails). However, the algorithm samples with replacement, making this event extremely unlikely.

---

**Lemma 2** is more reasonable. It states that the expected energy is correct. Some notational improvements can be made in equation (13):
- You can remove the factor $\frac{1}{|\mathcal{K}|}$ and the summation over $T_k \in \mathcal{K}$, and simply replace $T_k$ with $T$. It suffices to show that the expectation over a single random spanning tree $T\sim \Omega(\mathcal{T})$ is correct. Since $\mathcal{K}$ consists of i.i.d. samples, the expectation of their average is also correct.
- If you prefer to keep the set $\mathcal{K}$, then the expectation should be taken over the randomness of all i.i.d. samples in $\mathcal{K}$, where each $T_k \sim \Omega(\mathcal{T})$.

---

> ### Author Response · Authors · 2025-11-21
>
> Thank you for your valuable feedback regarding content clarity and writing quality.
>
> > **Q1/W1.**
>
> Your question captures the key point when we use BP to solve problems on MRFs. In general, using the BP algorithm is like using dynamic programming: we solve the problem by accessing nodes in a certain order. BP is more general because it can work on arbitrary graphs rather than only on trees. For reasons of efficiency and accuracy, BP and BP-based algorithms are among the most popular methods for MRF problems.
>
> Finding the global minimum on a general graph by message passing is intractable, so our practical approach is to **first run belief propagation to incorporate global information into the node marginals, and then sample from these marginals to obtain a solution. It is a general method and details can be found in [1].**
>
> There is considerable prior work that tries to merge inference results from subgraphs to overcome BP’s inexactness on general graphs, often using naive schemes such as voting. These approaches perform poorly in our experiments (see Appendix G, where we compare to the method in [1]). The main problem is that they ignore the bias between substructures and the original MRF. Our contribution is to **use a reweighting mechanism that compensates for this bias so that merging BP results from trees better approximates inference on the full graph.** This preserves desirable BP properties (efficiency and exactness on trees) while improving inference quality.
>
> **The example you gave is essentially the idea behind our algorithm: we obtain marginals and then make node-wise selections.** Because each marginal already incorporates information from the rest of the graph, the selection at each node implicitly accounts for other nodes. Symmetry (multiple equal-energy solutions) can be troublesome in practice, though it is rare for complex MRFs where nodes have different value sets. This issue is manageable: we can re-sample multiple times to find better solutions, and when multiple selections share the same energy, any of them is acceptable for our objectives.
>
> Our algorithm is designed for general MRFs, but—as shown in the experiments—it particularly excels on MRFs with grid- or cell-like structures where standard BP is negatively affected by the numerous loops.
>
> [1] Firas Hamze and Nando de Freitas. From fields to trees. In Proceedings of the 20th Conference on Uncertainty in Artificial Intelligence, pp. 243–250, 2004.
>
> > **lemma 1/lemma 2**
>
> Thank you for the insightful question. The intuition behind Lemma 1 is that, in case where we could obtain all unique spanning trees, our approximation would coincide exactly with the original BP. Since collecting all spanning trees is impractical, we implement the idea using an algorithm based on tree sampling. We provide a theoretical error bound for our algorithm to quantify this gap.
>
> Your suggestions on the expressions in Lemma 1 and Lemma 2 were very helpful. We have updated the PDF and revised the lemma statements as you recommended.

---

> ### Comment · Reviewer_14zd · 2025-11-22
> **Reply**
>
> Thank you for your clarification. My overall evaluation remains the same.

---

> > ### Author Response · Authors · 2025-11-23
> >
> > Thank you for taking the time to carefully review our responses. We sincerely appreciate your continued positive assessment and your support for our work. Your thoughtful feedback has been invaluable in helping us refine our paper, and we are grateful for your recommendation.

---

### Author Response · Authors · 2025-11-29

Dear ACs and reviewers,

We would like to express our sincere gratitude to all reviewers for their insightful suggestions and to the Area Chairs for their valuable work behind the scenes.

We appreciate the reviewers’ acknowledgment of our work’s contributions, specifically:

- The motivation is clear and the method is creative (all reviewers)
- We provide detailed instructions on how to implement our algorithm, which is easy to implement in itself (reviewers inkb and Vpu6)
- The empirical validation on synthetic + UAI + real PCI datasets shows promising gains for the proposed approach (reviewers 14zd and Vpu6)
- We provide a clear theoretical analysis of our algorithm (reviewer Vpu6)

We thank all the reviewers for their support regarding the acceptance of our work. All reviewers provided positive comments on our work’s originality, quality, and clarity, and reviewer inkb offered a particularly confident endorsement of its acceptance.

The reviewers' suggestions have significantly improved the quality of our paper. In response to their feedback, we have made the following enhancements:

- Added a new baseline, mapMAP, as suggested by reviewer inkb, which further demonstrates the superior performance of our algorithm.
- Added error bars to show the uncertainty of our algorithm and updated the results in Figure 1, as suggested by reviewer inkb, which illustrates the stability of our algorithm compared to the baselines.
- Fixed the typo in Lemma 2, as suggested by reviewer 14zd.

Best,
The authors

---

### Meta-Review · Area_Chair_EL5Y · 2025-12-16

**Summary:**

This paper addresses a problem of finding a MAP solution (corresponding to a configuration of states in minimum energy) in a MRF, which is a NP-hard in general. There are many approaches for approximating the solution on general cyclic graphs by utilizing tree-based decompositions or approximations. The method in this paper explicitly addresses the bias by reweighting edge potentials based on their eﬀective resistance, ensuring an unbiased approximation of the original energy function. The paper is well written and the idea is  simple and reasonable. It is properly justified by experiments.
Some of concerns raised by reviewers are summarized:
1. Why BP to compute the marginal distributions on trees instead of finding MAP solutions via DP on trees?
2. Some statement in the theorems and lemmas and some proofs look confusing.
3. The method relies on estimating the spanning three exactly requiring O(N^3) which limits the approach to small graphs and subtrees. While scalable approaches are discussed it is unclear how well they perform in practice.
4. The method is compared to very few competing methods.
5. In Theorem 1, the error bound decreases as the number of spanning trees increases. The proposed algorithm is taliored for sparse graphs. However, a sparse graph should have less spanning trees compared to a dense graph, which leads to a worse error bound. How do we understand this?

**Reviewer Concerns:**

All of concerns summarized above are properly addressed by authors. Regarding the more experiments, the authors did a good job during a limited period.

**Reviewer Scores:**

It follows from reviewers' earlier response to the rebuttal, the scores are expected to stay the same.

---

### Decision · Program_Chairs · 2026-01-26

Accept (Poster)